# Bridging Constraints and Stochasticity: A Fully First-Order Method for Stochastic Bilevel Optimization with Linear Constraints

## Abstract

This work provides the first finite-time convergence guarantees for linearly constrained stochastic bilevel optimization using only first-order methods, requiring solely gradient information without any Hessian computations or second-order derivatives. We address the unprecedented challenge of simultaneously handling linear constraints, stochastic noise, and finite-time analysis in bilevel optimization, a combination that has remained theoretically intractable until now. While existing approaches either require second-order information, handle only unconstrained stochastic problems, or provide merely asymptotic convergence results, our method achieves finite-time guarantees using gradient-based techniques alone. We develop a novel framework that constructs hypergradient approximations via smoothed penalty functions, using approximate primal and dual solutions to overcome the fundamental challenges posed by the interaction between linear constraints and stochastic noise. Our theoretical analysis provides explicit finite-time bounds on the bias and variance of the hypergradient estimator, demonstrating how approximation errors interact with stochastic perturbations. We prove that our first-order algorithm converges to $(\delta, \epsilon)$-Goldstein stationary points using $\Theta(\delta^{-1}\epsilon^{-5})$ stochastic gradient evaluations, establishing the first finite-time complexity result for this challenging problem class and representing a significant theoretical breakthrough in constrained stochastic bilevel optimization.

## 1 Introduction

Bilevel optimization is a powerful paradigm for hierarchical decision-making in machine learning, including hyperparameter tuning Franceschi et al. (2018), meta-learning Finn et al. (2017), and reinforcement learning Konda and Tsitsiklis (1999).

The standard formulation can be written as the following optimization problem:

$$\min_{x \in X} f(x, y^*(x)) \quad \text{s.t. } y^*(x) \in \arg\min_{y \in S(x)} g(x, y), \tag{1}$$

Here $S(x)$ denotes the feasible set of the lower-level problem (e.g., $S(x) = \mathbb{R}^m$ in the unconstrained case, or $S(x) = \{y : h(x, y) \leq 0\}$ for constrained cases). Furthermore, we define $F(x) := f(x, y^*(x))$ to be the overall bilevel objective as a function of $x$. Traditional methods of solving bilevel optimization often face scalability challenges Pedregosa (2016); Franceschi et al. (2017), including implicit differentiation with its requisite Hessian computations Amos and Kolter (2017); Ji and Yang (2021); Khanduri et al. (2024); Hu et al. (2023); Ghadimi and Wang (2018), and iterative differentiation characterized by high memory and computational demands. Shen et al. (2024); Brauer et al. (2024)

A recent breakthrough in bilevel optimization by Kwon et al. (2023); Liu et al. (2022) proposes using reformulation and penalty-based approaches to design a fully first-order gradient proxy. Several follow-up works based on this breakthrough have emerged, including improving finite time convergence of unconstrained bilevel optimization Chen et al. (2024a); Yang et al. (2023); Chen et al. (2024b); Kwon et al. (2024), constrained bilevel optimization Khanduri et al. (2023); Kornowski et al. (2024); Yao et al. (2024); Lu and Mei (2024), and applications of bilevel algorithms to machine learning Pan et al. (2024); Zhang et al. (2024); Petrulionytė et al. (2024).

However, many real-world scenarios involve *stochastic* objectives and *constraints* together, where gradients are noisy estimates from samples. While methods for unconstrained stochastic bilevel optimization have advanced (e.g., Ghadimi and Wang (2018); Kwon et al. (2023); Liu et al. (2022)), the confluence of stochasticity and explicit LL linear constraints poses significant unresolved challenges. A critical gap persists: no existing methods offer finite-time convergence guarantees for bilevel problems that are simultaneously *stochastic* and *linearly constrained* in the lower-level problem.

This paper bridges this gap by introducing the Fully First-order Constrained Stochastic Approximation (F2CSA) algorithm. We build upon the deterministic framework of Kornowski et al. (2024), developing a novel smoothed, stochastic hypergradient oracle tailored for bilevel problems with linearly constrained LL subproblems and stochastic objectives. The key to our approach is a smoothed reformulation that handles inexact dual variables, enabling robust hypergradient estimation from noisy first-order information and inexact LL solves. Our theoretical analysis, based on Lipschitz continuity and careful variance-bias tradeoffs, yields the first finite-time complexity guarantees for reaching $(\delta, \epsilon)$-Goldstein stationary points in this setting.

Our contributions include:

- **Stochastic Inexact Hypergradient Oracle:** We develop a stochastic inexact hypergradient oracle based on a smoothed Lagrangian method with penalty weights $\alpha_1 = \alpha^{-2}$ and $\alpha_2 = \alpha^{-4}$ where $\alpha > 0$. This oracle approximates hypergradients with bias bounded by $O(\alpha)$ and variance bounded by $O(1/N_g)$ using $N_g$ first-order gradient evaluations. Our smoothed Lagrangian method generalizes the approach from Kornowski et al. (2024) to allow approximate primal-dual lower-level solutions for constructing inexact hypergradient oracles.

- **Convergence Guarantees:** We apply the stochastic inexact hypergradient oracle with parameter $\alpha = \epsilon$ and $N_g = O(\epsilon^{-2})$ to design a double-loop algorithm for stochastic bilevel optimization problems with linear constraints. This algorithm attains a $(\delta, \epsilon)$-Goldstein stationary point of $F(x)$ with total first-order oracle complexity $\tilde{O}(\delta^{-1}\epsilon^{-5})$. This generalizes the deterministic bilevel optimization result with linear constraints (rate $\tilde{O}(\delta^{-1}\epsilon^{-4})$) to the stochastic setting.

Our work provides the first finite-time convergence guarantees for linearly constrained stochastic bilevel optimization under standard stochastic assumptions, providing a theoretically sound yet practical alternative to traditional bilevel optimization approaches.

## 2 RELATED WORK

**Penalty Methods and First-Order Reformulations.** To reduce the computational cost of second-order derivatives in bilevel optimization, recent works have proposed scalable, first-order algorithms based on penalty formulations Kwon et al. (2023); Liu et al. (2022). These techniques transform constrained bilevel problems into single-level surrogates that can be solved efficiently with convergence guarantees, where deterministic, partially stochastic, and fully stochastic bilevel optimization can achieve $\epsilon$-stationary point in $O(\epsilon^{-2})$, $O(\epsilon^{-4})$, $O(\epsilon^{-6})$ gradient calls, respectively Chen et al. (2024b). The convergence rate of the deterministic case can be further improved to $O(\epsilon^{-1.5})$ by momentum-based method Yang et al. (2023).

**Bilevel Optimization with Linear Constraints.** Due to the nonsmoothness of constrained bilevel optimization problems, Kornowski et al. (2024) focuses on Goldstein stationarity Goldstein (1977) and designs a new penalty method to achieve a zero-th order algorithm with $O(\delta^{-1}\epsilon^{-3})$ convergence and a first-order algorithm with $O(\delta^{-1}\epsilon^{-4})$ convergence to a $(\epsilon, \delta)$-Goldstein stationary point. On the other hand, Yao et al. (2024); Lu and Mei (2024) consider a different stationarity using $\epsilon$-KKT stationarity, where Lu and Mei (2024) achieves a $O(\epsilon^{-7})$ convergence rate, and Yao et al. (2024) achieves a $O(\epsilon^{-2})$ rate under a stronger assumption of access to projection operators. Compared to Goldstein stationarity, an $\epsilon$-KKT stationary point requires satisfying approximate KKT conditions (small constraint violation and small gradient norm of the Lagrangian), and hence is a stronger condition. We choose Goldstein's notion here as it naturally handles the nonsmoothness arising from the piecewise definition of $F(x)$ due to changing active constraints. (Notably, Yao et al. (2024) use a *doubly regularized gap function* approach and obtain fast rates to an $\epsilon$-KKT point under convexity

and with projection oracles; our method assumes strong convexity and LICQ but applies to nonconvex $F(x)$ and uses Goldstein's criterion.)

**Nonsmooth and nonconvex optimization.** The nonsmooth and nonconvex structure of bilevel optimization with constraints makes its analysis closely related to nonsmooth nonconvex optimization. The best-known convergence result is given by Zhang et al. (2020), which establishes optimal convergence rates of $O(\delta^{-1}\epsilon^{-3})$ for the deterministic case and $O(\delta^{-1}\epsilon^{-4})$ for the stochastic case. Our result of $\tilde{O}(\delta^{-1}\epsilon^{-5})$ is one factor of $\epsilon$ away from the optimal stochastic rate, indicating potential room for improvement. In particular, future work could explore using momentum (e.g., as in Yang et al. (2023)) or variance-reduction techniques to potentially improve the $\epsilon$-dependence.

## 3 Problem Formulation and Penalty-Based Approximation

We consider the following linearly constrained bilevel optimization problem:

$$\min_{x \in X} F(x) := \mathbb{E}_\xi \left[ f(x, y^*(x); \xi) \right]$$
$$\text{s.t.} \quad y^*(x) \in \arg\min_{y \in \mathbb{R}^m : h(x,y) \leq \mathbf{0}} \mathbb{E}_\zeta \left[ g(x, y; \zeta) \right] \tag{2}$$

Here, $x \in \mathbb{R}^n$ denotes the upper-level (UL) decision variable constrained to a closed convex set $X \subseteq \mathbb{R}^n$, and $y \in \mathbb{R}^m$ is the lower-level (LL) variable. The UL and LL objective functions $f(x, y; \xi)$ and $g(x, y; \zeta)$ are stochastic, depending on random variables $\xi$ and $\zeta$, respectively, which model data or simulation noise. Expectations are taken with respect to the underlying distributions of these random variables.

The LL feasible region is defined by a set of $p$ linear inequality constraints:

$$h(x, y) := \mathbf{A}x - \mathbf{B}y - \mathbf{b} \leq \mathbf{0}, \tag{3}$$

where $\mathbf{A} \in \mathbb{R}^{p \times n}$, $\mathbf{B} \in \mathbb{R}^{p \times m}$, and $\mathbf{b} \in \mathbb{R}^p$ are known matrices and vector. We assume the norm of the matrices are bounded by a given constant: $\|A\| \leq M_{\nabla h}$ and $\|B\| \leq M_{\nabla h}$, which also ensures that the Jacobian of the constraint function $h(x, y)$ is bounded.

Directly solving the stochastic bilevel problem is challenging due to the implicit dependence of $F(x)$ on $y^*(x)$ and the presence of noise in gradient and function evaluations.

### 3.1 Assumptions

We apply the following standard assumptions to our problem.

**Assumption 3.1** (Smoothness and Strong Convexity). *We make the following assumptions on the objectives $f, g$, constraints $h$, and associated matrices:*

  *(i) **Upper-Level Objective** $f$: The function $f(x, y)$ is $C_f$-smooth in $(x, y)$ (i.e., its gradient $\nabla f$ is $C_f$-Lipschitz continuous). The function $f(x, y)$ is also $L_f$-Lipschitz continuous in $(x, y)$.*

  *Note: $L_f$-Lipschitz continuity of $f$ implies that its gradient norm is bounded, i.e., $\|\nabla f(x, y)\| \leq L_f$.*

  *(ii) **Lower-Level Objective** $g$: The function $g(x, y)$ is $C_g$-smooth in $(x, y)$ (i.e., its gradient $\nabla g$ is $C_g$-Lipschitz continuous). For each fixed $x \in X$, the function $g(x, \cdot)$ is $\mu_g$-strongly convex in $y$, with $\mu_g > 0$. The gradient norm is bounded: $\|\nabla g(x, y)\| \leq L_g$. This strong convexity ensures a unique LL minimizer $y^*(x)$ for each $x$, necessary for our hypergradient formulation. The bounded gradient assumption is standard in stochastic optimization.*

  *(iii) **Constraint Qualification (LICQ)**: The Linear Independence Constraint Qualification holds for the lower-level constraints at the optimal solution $y^*(x)$ for all $x \in X$. (Specifically, the Jacobian of the active constraints with respect to $y$, given by $-\mathbf{B}$ restricted to its active rows, has full row rank.)*

Under Assumption 3.1, the uniqueness of the LL solution $y^*(x)$ and multipliers $\lambda^*(x)$ is guaranteed. LICQ can potentially be relaxed to weaker qualifications at the cost of more complex analysis; we impose LICQ for simplicity.

**Assumption 3.2** (Global Lipschitz continuity of LL solution). *The optimal lower-level solution map $y^*(x)$ is globally $L_y$-Lipschitz continuous in $x$ on $X$. That is, there exists $L_y \geq 0$ such that $\|y^*(x) - y^*(x')\| \leq L_y\|x - x'\|$ for all $x, x' \in X$.*

This assumption is standard in bilevel optimization (Facchinei and Pang, 2003). Given global Lipschitzness, the UL objective $F(x) = f(x, y^*(x))$ is $L_F$-Lipschitz continuous with $L_F \leq L_{f,x} + L_{f,y}L_y$. This ensures $y^*(x)$ varies Lipschitzly with $x$, which we use to control errors in our convergence analysis.

We assume access only to noisy first-order information via stochastic oracles.

**Assumption 3.3** (Stochastic Oracles). *Stochastic first-order oracles (SFOs) $\nabla \tilde{f}(x, y, \xi)$, $\nabla \tilde{g}(x, y; \zeta)$ are available, satisfying:*

(i) ***Unbiasedness:*** $\mathbb{E}[\nabla \tilde{f}(x, y; \xi)] = \nabla f(x, y)$ *and* $\mathbb{E}[\nabla \tilde{g}(x, y; \zeta)] = \nabla g(x, y)$.

(ii) ***Bounded Variance:*** $\mathbb{E}[\|\nabla \tilde{f} - \nabla f\|^2] \leq \sigma^2$ *and* $\mathbb{E}[\|\nabla \tilde{g} - \nabla g\|^2] \leq \sigma^2$.

Assumptions 3.1–3.3 are standard in bilevel optimization and provide the necessary smoothness, convexity, and stability conditions for our analysis.

## 3.2 GOLDSTEIN STATIONARITY

Due to the potential nonsmoothness of $F(x)$, we target Goldstein stationarity, a robust concept for nonsmooth optimization.

**Definition 3.1** (Goldstein Subdifferential (Goldstein, 1977)). *For an $L_F$-Lipschitz function $F : \mathbb{R}^n \rightarrow \mathbb{R}$, $x \in \mathbb{R}^n$, $\delta > 0$:*

$$\partial_\delta F(x) := \text{conv}\left(\bigcup_{z \in B_\delta(x)} \partial F(z)\right)$$

*where $\partial F(z)$ is the Clarke subdifferential, and $B_\delta(x)$ is the $\delta$-ball around $x$.*

**Definition 3.2** ($(\delta, \epsilon)$-Goldstein Stationarity). *A point $x \in X$ is $(\delta, \epsilon)$-Goldstein stationary if*

$$dist(\mathbf{0}, \partial_\delta F(x) + \mathcal{N}_X(x)) \leq \epsilon,$$

*where $\mathcal{N}_X(x)$ is the normal cone to $X$ at $x$.*

## 4 ERROR ANALYSIS FOR STOCHASTIC HYPERGRADIENT

We first control the effect of inexact dual variables on the penalty gradient and propagate this control to the shift of the penalized lower-level minimizer (Lemmas 4.1 and 4.2), which together yield an $O(\alpha)$ bias for the oracle (Lemma 4.3). We then bound the sampling variance by $O(1/N_g)$ (Lemma 4.4); Theorem 4.1 consolidates these bounds, and Lemma 4.5 records the inner method cost $\tilde{O}(\alpha^{-2})$.

**Notation:** We use tildes for stochastic/inexact quantities: $\tilde{y}^*(x)$, $\tilde{\lambda}(x)$ (approximate LL primal-dual solutions), $\tilde{y}(x)$ (approximate penalized minimizer), and $\nabla \tilde{F}(x)$ (stochastic hypergradient estimator). True quantities lack tildes: $y^*(x)$, $\lambda^*(x)$, and $\nabla F(x)$. The estimator $\nabla \tilde{F}(x) = \nabla_x L_{\tilde{\lambda},\alpha}(x, \tilde{y}(x))$ approximates $\nabla F(x)$ with bias $O(\alpha)$ and variance $O(1/N_g)$ (Lemmas 4.3 and 4.4).

## 4.1 STOCHASTIC IMPLEMENTATION

We compute the stochastic hypergradient oracle via a penalty formulation with smooth activation as

**Remark 4.1** (Inner Loop Complexity). *The approximate LL solution $(\tilde{y}^*(x), \tilde{\lambda}(x))$ in Step 3 is obtained via a stochastic primal-dual method with updates: $y_{t+1} = y_t - \eta_y \nabla_y \tilde{g}(x, y_t; \zeta_t)$ and $\lambda_{t+1} = \max\{0, \lambda_t + \eta_\lambda(Ax - By_{t+1} - b)\}$. With $g(x, \cdot)$ strongly convex and smooth, this attains $O(\delta)$ accuracy in $O(\kappa_g \log(1/\delta))$ iterations (Lemma 4.5). Setting $\delta = \Theta(\alpha^3)$ and $\alpha = \Theta(\epsilon)$ yields $O(\kappa_g \log(1/\epsilon))$ iterations.*

**Algorithm 1** Stochastic Penalty-Based Hypergradient Oracle

---

1: **Input:** Point $x \in \mathbb{R}^n$, accuracy parameter $\alpha > 0$, variance bound $\sigma^2$ (bound on gradient noise variance per Assumption 3.3), batch size $N_g$

2: **Set:** $\alpha_1 = \alpha^{-2}$, $\alpha_2 = \alpha^{-4}$, $\delta = \Theta(\alpha^3)$

3: Compute approximate lower-level solution $(\tilde{y}^*(x), \tilde{\lambda}(x))$ by a stochastic primal-dual (SPD) method (see Lemma 4.5) such that $\|\tilde{y}^*(x) - y^*(x)\| \leq O(\delta)$ and $\|\tilde{\lambda}(x) - \lambda^*(x)\| \leq O(\delta)$, where $y^*(x)$ and $\lambda^*(x)$ denote the true lower-level minimizer and corresponding optimal multiplier

4: Define the smooth Lagrangian $L_{\tilde{\lambda},\alpha}(x,y)$ using the penalty Lagrangian (Eq. (4)) and smooth activation function $\rho(x)$ (defined below)

5: Compute $\tilde{y}(x) = \arg\min_y L_{\tilde{\lambda},\alpha}(x,y)$ by stochastic gradient steps such that $\|\tilde{y}(x) - y^*_{\tilde{\lambda},\alpha}(x)\| \leq \delta$, where $y^*_{\tilde{\lambda},\alpha}(x) := \arg\min_y L_{\tilde{\lambda},\alpha}(x,y)$

6: Collect $N_g$ i.i.d. samples $\{\xi_j\}_{j=1}^{N_g}$ and compute $\nabla \tilde{F}(x) = \frac{1}{N_g} \sum_{j=1}^{N_g} \nabla_x \tilde{L}_{\tilde{\lambda},\alpha}(x, \tilde{y}(x); \xi_j)$

7: **Output:** $\nabla \tilde{F}(x)$

---

**Smooth Activation Function:** To regularize constraint activation near the boundary, define $\rho_i(x) = \sigma_h(h_i(x, \tilde{y}^*(x))) \cdot \sigma_\lambda(\tilde{\lambda}_i(x))$ where:

$$\sigma_h(z) = \begin{cases} 0 & \text{if } z < -\tau\delta \\ \frac{\tau\delta + z}{\tau\delta} & \text{if } -\tau\delta \leq z < 0 \\ 1 & \text{if } z \geq 0 \end{cases}, \quad \sigma_\lambda(z) = \begin{cases} 0 & \text{if } z \leq 0 \\ \frac{z}{\epsilon_\lambda} & \text{if } 0 < z < \epsilon_\lambda \\ 1 & \text{if } z \geq \epsilon_\lambda \end{cases}$$

with $\tau = \Theta(\delta)$ and $\epsilon_\lambda > 0$ being small positive parameters.

The penalty function for hypergradient estimation is:

$$L_{\tilde{\lambda},\alpha}(x,y) = f(x,y) + \alpha_1 \left( g(x,y) + (\tilde{\lambda}(x))^T h(x,y) - g(x, \tilde{y}^*(x)) \right) + \frac{\alpha_2}{2} \sum_{i=1}^{p} \rho_i(x) \cdot h_i(x,y)^2 \tag{4}$$

where $\alpha_1 = \alpha^{-2}$ and $\alpha_2 = \alpha^{-4}$ for $\alpha > 0$. The terms with $\tilde{\lambda}(x)$ and $\tilde{y}^*(x)$ promote KKT consistency and enforce constraints through a smoothed quadratic penalty.

The oracle outputs $\nabla \tilde{F}(x)$ with expectation $\mathbb{E}[\nabla \tilde{F}(x)] = \nabla_x L_{\tilde{\lambda},\alpha}(x, \tilde{y}(x))$. Its mean-squared error decomposes into bias and variance relative to $\nabla F(x)$:

$$\mathbb{E}[\|\nabla \tilde{F}(x) - \nabla F(x)\|^2] = \underbrace{\mathbb{E}[\|\nabla \tilde{F}(x) - \mathbb{E}[\nabla \tilde{F}(x)]\|^2]}_{\text{Variance}} + \underbrace{\|\mathbb{E}[\nabla \tilde{F}(x)] - \nabla F(x)\|^2}_{\text{Bias}^2} \tag{5}$$

We first bound the effect of using $\tilde{\lambda}(x)$ in place of $\lambda^*(x)$ on the gradient of the penalty Lagrangian.

**Lemma 4.1** (Lagrangian Gradient Approximation). *Assume $\|\tilde{\lambda}(x) - \lambda^*(x)\| \leq C_\lambda \delta$ and under Assumption 3.1 (iii), let $\alpha_1 = \alpha^{-2}$, $\alpha_2 = \alpha^{-4}$, and $\tau = \Theta(\delta)$. Then for fixed $(x,y)$:*

$$\|\nabla L_{\lambda^*,\alpha}(x,y) - \nabla L_{\tilde{\lambda},\alpha}(x,y)\| \leq O(\alpha_1\delta + \alpha_2\delta).$$

*Proof sketch.* Consider $\Delta\lambda := \lambda^*(x) - \tilde{\lambda}(x)$ and decompose the gradient difference into a linear penalty part and a quadratic penalty part. For the linear term, $\|\nabla h\|$ is bounded by Assumption 3.1(iii) and $\|\Delta\lambda\| \leq C_\lambda\delta$, yielding $O(\alpha_1\delta)$. For the quadratic term, only near-active constraints contribute where $|h_i(x, \tilde{y}^*(x))| \leq \tau\delta$, and one obtains two pieces: $\alpha_2 \sum_i (\rho_i^* - \tilde{\rho}_i) h_i \nabla h_i$ and $\frac{\alpha_2}{2} \sum_i h_i^2 \nabla(\rho_i^* - \tilde{\rho}_i)$. Using $|h_i| = O(\delta)$ in these regions, $\|\nabla h_i\|$ bounded, and $\|\nabla(\rho_i^* - \tilde{\rho}_i)\| = O(1/\delta)$, both pieces are $O(\alpha_2\delta)$. Combining yields $O(\alpha_1\delta + \alpha_2\delta)$.

Building on this result, we next bound the difference between exact solutions for true and approximate dual variables:

**Lemma 4.2** (Solution Error). *Let $y_{\lambda,\alpha}^*(x) := \arg\min_y L_{\lambda,\alpha}(x,y)$ with $\alpha_1 = \alpha^{-2}$ and $\alpha_2 = \alpha^{-4}$.*

*Assume the target accuracy parameter $\alpha$ is small enough that $\mu_{pen} = \alpha_1 \mu_g - \frac{1}{2}C_f > 0$, where $C_f$ is the smoothness constant of $f$ and $\mu_g$ is the strong convexity constant of $g(x,\cdot)$ as per Assumption 3.1. This ensures $\mu_{pen} \geq \frac{1}{2}\alpha_1 \mu_g$, so that $L_{\lambda,\alpha}(x,y)$ is $\mu = \Omega(\alpha \mu_g)$-strongly convex in $y$.*

*If the dual approximation satisfies $\|\tilde{\lambda}(x) - \lambda^*(x)\| \leq C_\lambda \delta$ and the gradient bound from Lemma 4.1 holds, then:*

$$\|y_{\lambda^*,\alpha}^*(x) - y_{\tilde{\lambda},\alpha}^*(x)\| \leq \frac{C_{sol}}{\mu}(\alpha_1 + \alpha_2)\delta,$$

*where the constant $C_{sol}$ depends on $C_\lambda$ and $M_{\nabla h}$ (Assumption 3.1(iii) on $\|\nabla h\|$ bound).*

Strong convexity together with Lemma 4.1 yields the stated solution bound.

With controlled approximation errors, we now derive a systematic bias bound.

## 4.2 BIAS ANALYSIS (DETERMINISTIC ERROR)

The bias is the deterministic error $\mathbb{E}[\nabla \tilde{F}(x)] - \nabla F(x)$, due to the penalty surrogate and the use of inexact inner solutions $(\tilde{\lambda}, \tilde{y})$ in place of $(\lambda^*, y^*, y_{\lambda^*,\alpha}^*)$.

**Lemma 4.3** (Hypergradient Bias Bound). *Let $\nabla_x L_{\lambda,\alpha}(x,y)$ denote the partial gradient of the penalty Lagrangian with respect to $x$. Assume it is $L_{H,y}$-Lipschitz in $y$ and $L_{H,\lambda}$-Lipschitz in $\lambda$. With $\alpha_1 = \alpha^{-2}$, $\alpha_2 = \alpha^{-4}$, choose $\delta = \Theta(\alpha^3)$ and suppose $\|\tilde{y}(x) - y_{\tilde{\lambda},\alpha}^*(x)\| \leq \delta$ and $\|\tilde{\lambda}(x) - \lambda^*(x)\| \leq C_\lambda \delta$. If $L_{\lambda^*,\alpha}(x,\cdot)$ is $\mu$-strongly convex with $\mu \geq c_\mu \alpha^{-2}$, then*

$$\|\mathbb{E}[\nabla \tilde{F}(x)] - \nabla F(x)\| \leq C_{bias}\alpha,$$

*where $C_{bias}$ depends only on $L_{H,y}$, $L_{H,\lambda}$, $C_g$, $C_\lambda$, $c_\mu$, and the penalty constant $C_{pen}$. Here $L_{H,y}$ and $L_{H,\lambda}$ are the Lipschitz constants of $\nabla_x L_{\lambda,\alpha}(x,y)$ with respect to $y$ and $\lambda$ (from the lemma assumptions); $C_g$ is the Lipschitz constant of $\nabla_y g$ (from Assumption 3.1(ii)); $C_\lambda$ is an upper bound on $\|\lambda^*(x)\|$ (guaranteed by strong convexity and LICQ); $c_\mu$ is a positive constant linking the lower-level strong convexity to $\alpha^{-2}$ (see Lemma 4.2); and $C_{pen}$ is the penalty parameter in our formulation (determined by the choice of $\alpha_1$ and $\alpha_2$ sufficiently large such that the penalty term dominates any curvature of $f$).*

*Proof sketch.* By the triangle inequality, write $\|\mathbb{E}[\nabla \tilde{F}(x)] - \nabla F(x)\| \leq T_1 + T_2 + T_3$, where $T_1 = \|\nabla_x L_{\tilde{\lambda},\alpha}(x,\tilde{y}(x)) - \nabla_x L_{\tilde{\lambda},\alpha}(x,y_{\tilde{\lambda},\alpha}^*(x))\| \leq L_{H,y}\,\delta = O(\alpha^3)$ for $\delta = \Theta(\alpha^3)$; $T_2 = \|\nabla_x L_{\tilde{\lambda},\alpha}(x,y_{\tilde{\lambda},\alpha}^*(x)) - \nabla_x L_{\lambda^*,\alpha}(x,y_{\lambda^*,\alpha}^*(x))\| \leq L_{H,\lambda}C_\lambda\,\delta + L_{H,y}\,\|y_{\tilde{\lambda},\alpha}^*(x) - y_{\lambda^*,\alpha}^*(x)\|$ and Lemma 4.2 gives $\|y_{\tilde{\lambda},\alpha}^* - y_{\lambda^*,\alpha}^*\| \leq (C_{sol}/\mu)(\alpha_1 + \alpha_2)\delta$, so with $\alpha_1 = \alpha^{-2}$, $\alpha_2 = \alpha^{-4}$, $\delta = \Theta(\alpha^3)$, $\mu = \Theta(\alpha^{-2})$ we obtain $T_2 = O(\alpha)$; $T_3 = \|\nabla_x L_{\lambda^*,\alpha}(x,y_{\lambda^*,\alpha}^*(x)) - \nabla F(x)\| = O(C_{pen}\alpha)$ by Kornowski et al. (2024). Hence $\|\mathbb{E}[\nabla \tilde{F}(x)] - \nabla F(x)\| = O(\alpha)$. $\qquad\square$

Thus the bias scales as $O(\alpha)$ when the inner accuracy is set to $\delta = \Theta(\alpha^3)$. This means that $\nabla \tilde{F}(x)$ is an $\alpha$-accurate estimator of $\nabla F(x)$ in expectation.

## 4.3 VARIANCE ANALYSIS (STOCHASTIC ERROR)

The variance, $\mathrm{Var}_{x,\tilde{\lambda},\tilde{y}}(\nabla \tilde{F}(x)) = \mathbb{E}_{x,\tilde{\lambda},\tilde{y}}[\|\nabla \tilde{F}(x) - \mathbb{E}[\nabla \tilde{F}(x)]\|^2]$, quantifies the error due to a finite batch size $N_g$ in estimating $\mathbb{E}[\nabla \tilde{F}(x)]$.

**Lemma 4.4** (Variance Bound). *Under Assumption 3.3 (i)–(ii), let $\sigma^2$ be a uniform bound on*

$\mathrm{Var}_{x,\tilde{\lambda},\tilde{y}}(\nabla_x \tilde{L}_{\tilde{\lambda},\alpha}(x,\tilde{y};\xi))$.

*With a mini-batch of $N_g$ i.i.d. samples in Algorithm 1, the conditional variance of the hypergradient estimate satisfies*

$$\mathrm{Var}_{x,\tilde{\lambda},\tilde{y}}(\nabla \tilde{F}(x)) \leq \frac{\sigma^2}{N_g}.$$

*Proof Sketch.* The hypergradient estimate $\nabla \tilde{F}(x)$ is the average of $N_g$ i.i.d. random vectors $G_j = \nabla_x \tilde{L}_{\tilde{\lambda}, \alpha}(x, \tilde{y}; \xi_j)$.

By Assumption 3.3(ii), each term $G_j$ has conditional variance bounded by $\sigma^2$, i.e.,

$\mathrm{Var}_{x, \tilde{\lambda}, \tilde{y}}(G_j) \leq \sigma^2$. Since the samples $\xi_j$ are i.i.d., the terms $G_j$ are conditionally independent, and the conditional variance of their average is bounded accordingly. This follows standard mini-batch averaging analysis. $\qquad\square$

### 4.4 COMBINED ERROR BOUNDS

We combine the bias and variance bounds to characterize the overall accuracy of the hypergradient oracle.

**Theorem 4.1** (Accuracy of Stochastic Hypergradient). *Let $\nabla \tilde{F}(x)$ be the output of Algorithm 1 with penalty parameters $\alpha_1 = \alpha^{-2}, \alpha_2 = \alpha^{-4}$, and inner accuracy $\delta = O(\alpha^3)$. There exists a constant $C_{bias}$ such that:*

$$\mathbb{E}[\|\nabla \tilde{F}(x) - \nabla F(x)\|^2] \leq 2C_{bias}^2 \alpha^2 + \frac{2\sigma^2}{N_g}.$$

*Proof Sketch.* We apply the standard bias–variance decomposition. The bias term is bounded using Lemma 4.3, which shows that the expected output of the oracle approximates the true gradient up to an $O(\alpha)$ error. - The variance term is controlled using Lemma 4.4, which shows that averaging $N_g$ noisy gradients leads to variance bounded by $\sigma^2/N_g$.

Adding these two contributions and applying Jensen's inequality yields the desired total error bound. $\qquad\square$

**Lemma 4.5** (Inner-loop Oracle Complexity). *Fix $\alpha > 0$ and set $\alpha_1 = \alpha^{-2}$, $\alpha_2 = \alpha^{-4}$, $\delta = \Theta(\alpha^3)$. Let $g(x, \cdot)$ be $\mu_g$-strongly convex and $C_g$-smooth, and the stochastic oracles of Assumption 3.3 have variance $\sigma^2$. Choose the mini-batch size $N_g = \sigma^2/\alpha^2$. Running Algorithm 1 with $\tilde{O}(\alpha^{-2})$ stochastic first-order oracle (SFO) calls in its inner loops yields a stochastic inexact gradient $\nabla \tilde{F}(x)$ characterized by bias of $O(\alpha)$ and variance of $O(\alpha^2)$.*

## 5 STOCHASTIC BILEVEL ALGORITHM AND CONVERGENCE ANALYSIS

We now introduce the principal algorithm, F2CSA (Algorithm 2), which leverages the previously analyzed stochastic hypergradient oracle within a non-smooth, non-convex optimization framework.

We then present detailed convergence proofs that provide rigorous guarantees for identifying $(\delta, \epsilon)$-Goldstein stationary points.

Algorithm 2 provides an iterative framework leveraging our inexact stochastic hypergradient oracle. The method maintains a direction term $\Delta_t$, updated using a momentum-like step involving the oracle's output $g_t = \nabla \tilde{F}(z_t)$ and subsequently clipped to ensure $\|\Delta_t\| \leq D$. The output iterates $x_k$ are constructed by averaging sample points $z_t$ to approximate the Goldstein subdifferential Goldstein (1977); Zhang et al. (2020); Davis and Drusvyatskiy (2019).

**Remark 5.1** (Integration with Stochastic Hypergradient Oracle). *Algorithm 2 uses Algorithm 1 as its gradient estimation subroutine. We fix $\alpha = \Theta(\epsilon)$, $N_g = \Theta(\sigma^2/\alpha^2)$, and $\delta = \Theta(\alpha^3)$ as constants (set as functions of $\epsilon$ at the outset) rather than using time-decaying schedules. The step size $\eta = \Theta(\delta\epsilon^3)$ is constant (Theorem 5.1); in practice we tune $\eta$ but do not decay it due to clipping.*

### 5.1 CONVERGENCE TO GOLDSTEIN STATIONARITY

The following theorem establishes convergence to $(\delta, \epsilon)$-Goldstein stationarity (Definition 3.2) with an inexact gradient oracle having bounded error.

**Theorem 5.1** (Convergence with Stochastic Hypergradient Oracle). *Suppose $F : \mathbb{R}^n \to \mathbb{R}$ is $L_F$-Lipschitz. Let $\nabla \tilde{F}(\cdot)$ be a stochastic hypergradient oracle satisfying:*

**Algorithm 2** Nonsmooth Nonconvex Algorithm with Inexact Stochastic Hypergradient Oracle

---

1: **Input:** Initialization $x_0 \in \mathbb{R}^n$, clipping parameter $D > 0$, step size $\eta > 0$, Goldstein accuracy $\delta > 0$, iteration budget $T \in \mathbb{N}$, inexact stochastic gradient oracle $\nabla \tilde{F} : \mathbb{R}^n \to \mathbb{R}^n$
2: **Initialize:** $\Delta_1 = 0$
3: **for** $t = 1, \dots, T$ **do**
4:     Sample $s_t \sim \text{Unif}[0, 1]$
5:     $x_t = x_{t-1} + \Delta_t$, $z_t = x_{t-1} + s_t \Delta_t$
6:     Compute $g_t = \nabla \tilde{F}(z_t)$ by running Algorithm 1 with $N_g = \Theta(\sigma^2/\alpha^2)$ samples, so the inexact gradient has bias $O(\alpha)$ and variance $O(\alpha^2)$.
7:     $\Delta_{t+1} = \text{clip}_D(\Delta_t - \eta g_t)$ $\{\text{clip}_D(v) := \min\{1, \frac{D}{\|v\|}\} \cdot v\}$
8: **end for**
9: $M = \lfloor \frac{\delta}{D} \rfloor$, $K = \lfloor \frac{T}{M} \rfloor$ $\{\text{Group iterations for Goldstein subdifferential}\}$
10: **for** $k = 1, \dots, K$ **do**
11:     $x_k = \frac{1}{M} \sum_{m=1}^{M} z_{(k-1)M+m}$
12: **end for**
13: **Output:** $x_{\text{out}} \sim \text{Uniform}\{x_1, \dots, x_K\}$

---

    1. *Bias bound:* $\|\mathbb{E}[\nabla \tilde{F}(x)] - \nabla F(x)\| \le C_{bias}\alpha$

    2. *Variance bound:* $\mathbb{E}[\|\nabla \tilde{F}(x) - \mathbb{E}[\nabla \tilde{F}(x)]\|^2] \le \frac{\sigma^2}{N_g}$

*Then running Algorithm 2 with parameters $D = \Theta(\frac{\delta\epsilon^2}{L_F^2})$, $\eta = \Theta(\frac{\delta\epsilon^3}{L_F^4})$, and $N_g = \Theta(\frac{\sigma^2}{\alpha^2})$ outputs a point $x_{out}$ such that $\mathbb{E}[dist(\mathbf{0}, \partial_\delta F(x_{out}))] \le \epsilon + O(\alpha)$, using $T = O(\frac{(F(x_0)-\inf F)L_F^2}{\delta\epsilon^3})$ calls to $\nabla \tilde{F}(\cdot)$.*

*Proof Sketch.* We first take $N_g = \Theta(\sigma^2/\alpha^2)$ to get bias $O(\alpha)$ and variance $O(\alpha^2)$.

Clip online gradient descent telescopes; error terms are $O(\eta)$ (stability) and $O(D\alpha)$ (stochastic).

Choose $M = \Theta(\epsilon^{-2})$ and $D = \Theta(\delta\epsilon^2)$ so $\|z_t - x_k\| \le MD \le \delta$ and the block average lies in $\partial_\delta F(x_k)$.

Set $\eta = \Theta(\delta\epsilon^3)$ and run $T = O(((F(x_0) - \inf F)L_F^2)/(\delta\epsilon^3))$ to obtain $\mathbb{E}[dist(\mathbf{0}, \partial_\delta F(x_{\text{out}}))] \le \epsilon + O(\alpha)$.

Finally, set $\alpha = \Theta(\epsilon)$ to conclude $\Theta(\epsilon)$ stationarity. $\qquad\qquad\square$

Using Theorem 5.1, we can finally show the overall complexity of stochastic constrained bilevel optimization.

**Theorem 5.2** (Complexity of solving stochastic constrained bilevel optimization)**.** *The total stochastic first-order oracle (SFO) complexity is*

$$T \cdot N_g = \Theta\left(\frac{F(x_0) - \inf F}{\delta\epsilon^3}\right) \cdot \Theta\left(\frac{\sigma^2}{\epsilon^2}\right) = \Theta\left(\frac{(F(x_0) - \inf F)\sigma^2}{\delta\epsilon^5}\right) \tag{6}$$

*Including logarithmic factors from the inner loops, this becomes:*

$$SFO\ complexity = \tilde{O}\left(\frac{(F(x_0) - \inf F)\sigma^2}{\delta\epsilon^5}\right) = \tilde{O}(\delta^{-1}\epsilon^{-5}) \tag{7}$$

## 6   EXPERIMENTS

To validate our theoretical analysis and assess the practical performance of the proposed F2CSA algorithm, we conduct experiments on synthetic bilevel optimization problems. We compare our method against SSIGD Khanduri et al. (2023) and DSBLO Khanduri et al. (2024), both Hessian-based approaches by Khanduri et al. SSIGD uses an implicit gradient approach while DSBLO employs a doubly stochastic bilevel method. These comparisons highlight the computational advantages of our first-order approach over methods requiring second-order information.

## 6.1 PROBLEM SETUP

We evaluate our approach on toy bilevel problems with box constraints:

$$\min_{x \in \mathbb{R}^d} f(x, y^*(x)) := \tfrac{1}{2} x^\top Q_u x + c_u^\top x + \tfrac{1}{2} y^\top P y + x^\top P y \tag{8}$$

$$\text{s.t. } y^*(x) \in \arg\min_{y \in [-\mathbf{1}, \mathbf{1}]} g(x, y) := \tfrac{1}{2} y^\top Q_l y + c_l^\top y + x^\top y \tag{9}$$

Parameters $Q_u, Q_l, P, c_u, c_l$ are sampled from zero-mean Gaussians. Stochasticity is introduced by adding Gaussian noise $\mathcal{N}(0, \sigma^2)$ to the quadratic terms during gradient evaluations with noise standard deviation $\sigma = 0.01$. All algorithms use identical problem instances, initial points, random seeds, and the same lower-level solver to ensure fair evaluation. Step sizes are calibrated to be comparable across methods: SSIGD employs diminishing step sizes with $\beta = 10^{-4}$, DSBLO uses adaptive step size selection, and F2CSA utilizes fixed step size $\eta = 10^{-5}$, reflecting their different algorithmic structures.

## 6.2 RESULTS AND ANALYSIS

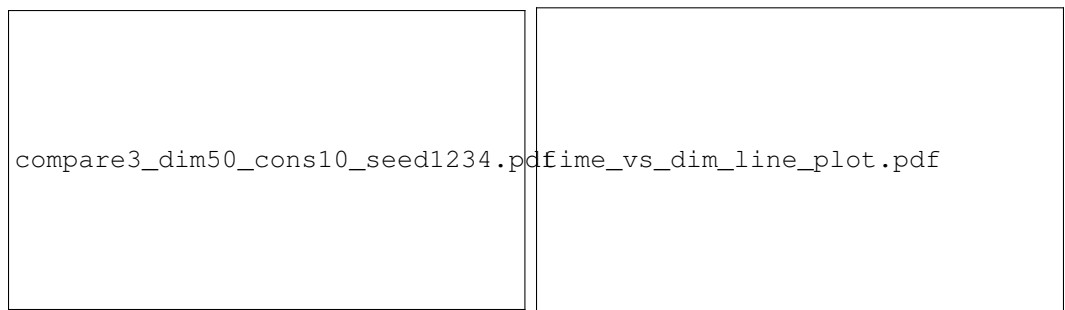

Figure 1: Loss convergence trajectories for F2CSA, SSIGD, and DSBLO in dimension 50.

Figure 2: Computational cost scaling with problem dimension.

### CONVERGENCE PERFORMANCE

Figure 1 shows convergence trajectories on a 50-dimensional problem. All three methods converge to similar final loss values, with F2CSA maintaining stable convergence. The comparable performance of F2CSA to Hessian-based methods is consistent with our theoretical analysis in Lemma 4.3, which bounds the oracle error at $O(\alpha)$ bias and $O(1/N_g)$ variance.

### COMPUTATIONAL SCALABILITY

Figure 2 shows computational cost scaling with problem dimension from $d = 100$ to $d = 4000$. The plot reveals a crossover point around $d = 1000$: for $d < 1000$, Hessian-based methods (DSBLO and SSIGD) are faster, while for $d > 1000$, F2CSA becomes increasingly advantageous. At $d = 4000$, F2CSA requires 7.7 seconds compared to 22.6 seconds for DSBLO and 22.0 seconds for SSIGD, representing approximately $3\times$ speedup. The plot shows F2CSA maintains near-linear growth, while Hessian-based methods exhibit super-linear growth as dimension increases.

### KEY INSIGHTS

The experimental results demonstrate that F2CSA achieves comparable convergence performance to Hessian-based methods while providing superior computational efficiency in high dimensions. The crossover around $d = 1000$ and the $3\times$ speedup at $d = 4000$ validate the theoretical advantage of our first-order approach, which avoids quadratic-scaling Hessian computations. This makes F2CSA well-suited for high-dimensional applications where computational efficiency is critical.

## 7 CONCLUSION AND FUTURE WORK

We introduced a fully first-order framework for linearly constrained stochastic bilevel optimization and established the first finite-time guarantee to $(\delta, \epsilon)$-Goldstein stationarity using a smoothed penalty-based hypergradient oracle. Section 4 quantified the oracle's error via an $O(\alpha)$ bias and $O(1/N_g)$ variance, which, together with the inner-loop cost in Lemma 4.5, yielded the calibrated choice $N_g = \Theta(\sigma^2/\alpha^2)$ and inner tolerance $\delta = \Theta(\alpha^3)$. Section 5 integrated this oracle into a clipped nonsmooth algorithm attaining $\mathbb{E}[\mathrm{dist}(0, \partial_\delta F(x_{\mathrm{out}}))] \leq \epsilon + O(\alpha)$ in $T = O(((F(x_0) - \inf F)L_F^2)/(\delta\epsilon^3))$ iterations; setting $\alpha = \Theta(\epsilon)$ implies the overall SFO complexity $\tilde{O}(\delta^{-1}\epsilon^{-5})$. Experiments corroborated the theory: F2CSA scales favorably in high dimensions, trading a small loss gap for speed, and outperforms Hessian-based baselines in wall-clock time at large $d$ without sacrificing solution quality.

Two limitations are noteworthy. First, the rate is one factor of $\epsilon$ from the best-known stochastic nonsmooth complexity, suggesting headroom for variance reduction or momentum. Second, our analysis hinges on LICQ, strong convexity of the lower level, and linear constraints; relaxing these raises technical challenges. (LICQ could potentially be relaxed to weaker conditions at the cost of more complex analysis.)

Promising directions include: (i) variance-reduced estimators or momentum to approach $\tilde{O}(\delta^{-1}\epsilon^{-4})$; (ii) structure-aware penalties stable under weaker qualifications; (iii) specialized treatments of one-sided stochasticity; and (iv) extending to nonlinear constraints. These would broaden practicality in meta-learning, RL, and large-scale ERM scenarios.

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

# A  APPENDIX

**Lemma 4.1** (Lagrangian Gradient Approximation). *Assume* $\|\tilde{\lambda}(x) - \lambda^*(x)\| \leq C_\lambda \delta$ *and under Assumption 3.1 (iii), let* $\alpha_1 = \alpha^{-2}$, $\alpha_2 = \alpha^{-4}$, *and* $\tau = \Theta(\delta)$. *Then for fixed* $(x, y)$:
$$\|\nabla L_{\lambda^*,\alpha}(x,y) - \nabla L_{\tilde{\lambda},\alpha}(x,y)\| \leq O(\alpha_1 \delta + \alpha_2 \delta).$$

*Proof.* Define penalty Lagrangian:
$$L_{\lambda,\alpha}(x,y) = f(x,y) + \alpha_1(g(x,y) + \lambda^T h(x,y) - g(x,\tilde{y}^*(x))) \tag{10}$$
$$+ \frac{\alpha_2}{2} \sum_{i=1}^{p} \rho_i(x) h_i(x,y)^2 \tag{11}$$

with activation $\rho_i(x) = \sigma_h(h_i(x,\tilde{y}^*(x))) \cdot \sigma_\lambda(\lambda_i(x))$ and dual error $\Delta\lambda = \lambda^*(x) - \tilde{\lambda}(x)$.

The gradient difference decomposes as:
$$\nabla L_{\lambda^*,\alpha} - \nabla L_{\tilde{\lambda}} = \underbrace{\alpha_1(\nabla h)^T \Delta\lambda}_{\text{Linear Term}} + \underbrace{\nabla\Delta_Q}_{\text{Quadratic Term}} \tag{12}$$

where $\Delta_Q = \frac{\alpha_2}{2} \sum_{i=1}^{p} \Delta\rho_i(x) h_i(x,y)^2$ with $\Delta\rho_i(x) = \rho_i^*(x) - \tilde{\rho}_i(x)$.

Linear Term in (12):

From $h(x,y) = \mathbf{A}x - \mathbf{B}y - \mathbf{b}$, we have:
$$\|\nabla h\| \leq \|\mathbf{A}\| + \|\mathbf{B}\| \leq 2M_{AB} \tag{13}$$

Using (13) and $\|\Delta\lambda\| \leq C_\lambda \delta$:
$$\|\alpha_1(\nabla h)^T \Delta\lambda\| \leq \alpha_1 \cdot 2M_{AB} \cdot C_\lambda \delta = O(\alpha_1 \delta) \tag{14}$$

Quadratic Term in (12):

The quadratic gradient expands to:
$$\nabla\Delta_Q = \underbrace{\alpha_2 \sum_{i=1}^{p} \Delta\rho_i h_i \nabla h_i}_{(T_1)} + \underbrace{\frac{\alpha_2}{2} \sum_{i=1}^{p} h_i^2 \nabla\Delta\rho_i}_{\text{T2}} \tag{15}$$

(T1): $\Delta\rho_i \neq 0$ only for near-active constraints where $|h_i(x,\tilde{y}^*(x))| \leq \tau\delta = O(\delta)$.

For constraint values:
$$h_i(x,y) - h_i(x,\tilde{y}^*(x)) = \mathbf{B}_i(\tilde{y}^*(x) - y) \tag{16}$$
$$\implies |h_i(x,y) - h_i(x,\tilde{y}^*(x))| \leq M_{AB}\delta \tag{17}$$

Using (17): $|h_i(x,y)| \leq O(\delta) + M_{AB}\delta = O(\delta)$ for near-active constraints.

With $|\Delta\rho_i| \leq 1$ and $\|\nabla h_i\| \leq 2M_{AB}$:
$$\|\Delta\rho_i h_i \nabla h_i\| \leq O(\delta) \implies \|\text{T1}\| \leq \alpha_2 p \cdot O(\delta) = O(\alpha_2 \delta) \tag{18}$$

$(T_2)$: $\|\nabla\Delta\rho_i\| = O(1/\delta)$ only when $|h_i(x,\tilde{y}^*(x))| = O(\delta)$.

In these regions: $|h_i(x,y)| = O(\delta)$ so:
$$h_i^2 \cdot \|\nabla\Delta\rho_i\| = O(\delta)^2 \cdot O(1/\delta) = O(\delta) \tag{19}$$

Using (19):
$$\|\text{T2}\| \leq \frac{\alpha_2}{2} p \cdot O(\delta) = O(\alpha_2 \delta) \tag{20}$$

From (14), (18), and (20):
$$\|\nabla L_{\lambda^*,\alpha} - \nabla L_{\tilde{\lambda}}\| \leq O(\alpha_1 \delta) + O(\alpha_2 \delta) + O(\alpha_2 \delta) = O(\alpha_1 \delta + \alpha_2 \delta) \tag{21}$$
$$\square$$

**Lemma 4.2** (Solution Error). *Let $y^*_{\lambda,\alpha}(x) := \arg\min_y L_{\lambda,\alpha}(x,y)$ with $\alpha_1 = \alpha^{-2}$ and $\alpha_2 = \alpha^{-4}$.*

*Assume the target accuracy parameter $\alpha$ is small enough that $\mu_{pen} = \alpha_1\mu_g - \frac{1}{2}C_f > 0$, where $C_f$ is the smoothness constant of $f$ and $\mu_g$ is the strong convexity constant of $g(x,\cdot)$ as per Assumption 3.1. This ensures $\mu_{pen} \geq \frac{1}{2}\alpha_1\mu_g$, so that $L_{\lambda,\alpha}(x,y)$ is $\mu = \Omega(\alpha\mu_g)$-strongly convex in $y$.*

*If the dual approximation satisfies $\|\tilde{\lambda}(x) - \lambda^*(x)\| \leq C_\lambda\delta$ and the gradient bound from Lemma 4.1 holds, then:*

$$\|y^*_{\lambda^*,\alpha}(x) - y^*_{\tilde{\lambda},\alpha}(x)\| \leq \frac{C_{sol}}{\mu}(\alpha_1 + \alpha_2)\delta,$$

*where the constant $C_{sol}$ depends on $C_\lambda$ and $M_{\nabla h}$ (Assumption 3.1(iii) on $\|\nabla h\|$ bound).*

*Proof.* For brevity, let $y^*_{\lambda^*,\alpha}(x) = y^*_{\lambda^*}$ and $y^*_{\tilde{\lambda},\alpha}(x) = y^*_{\tilde{\lambda}}$. From the definition of these minimizers, we have the first-order optimality conditions:

$$\nabla_y L_{\lambda^*,\alpha}(x, y^*_{\lambda^*}) = 0 \tag{22}$$
$$\nabla_y L_{\tilde{\lambda},\alpha}(x, y^*_{\tilde{\lambda}}) = 0 \tag{23}$$

The lemma assumes that $L_{\lambda,\alpha}(x,y)$ is $\mu$-strongly convex in $y$ (for relevant $\lambda$, including $\lambda^*$). Thus, $L_{\lambda^*,\alpha}(x,\cdot)$ is $\mu$-strongly convex. A standard property of a $\mu$-strongly convex function $\phi(y)$ with minimizer $y^*_1$ is that for any $y_2$:

$$\mu\|y^*_1 - y_2\|^2 \leq \langle \nabla_y\phi(y^*_1) - \nabla_y\phi(y_2), y^*_1 - y_2 \rangle$$

Applying this with $\phi(y) = L_{\lambda^*,\alpha}(x,y)$, $y^*_1 = y^*_{\lambda^*}$, and $y_2 = y^*_{\tilde{\lambda}}$:

$$\mu\|y^*_{\lambda^*} - y^*_{\tilde{\lambda}}\|^2 \leq \langle \nabla_y L_{\lambda^*,\alpha}(x, y^*_{\lambda^*}) - \nabla_y L_{\lambda^*,\alpha}(x, y^*_{\tilde{\lambda}}), y^*_{\lambda^*} - y^*_{\tilde{\lambda}} \rangle$$

Using the optimality condition from Eq. (22), $\nabla_y L_{\lambda^*,\alpha}(x, y^*_{\lambda^*}) = 0$, this simplifies to:

$$\mu\|y^*_{\lambda^*} - y^*_{\tilde{\lambda}}\|^2 \leq \langle -\nabla_y L_{\lambda^*,\alpha}(x, y^*_{\tilde{\lambda}}), y^*_{\lambda^*} - y^*_{\tilde{\lambda}} \rangle$$

Now, we add and subtract $\nabla_y L_{\tilde{\lambda},\alpha}(x, y^*_{\tilde{\lambda}})$ inside the inner product (and use $\nabla_y L_{\tilde{\lambda},\alpha}(x, y^*_{\tilde{\lambda}}) = 0$ from Eq. (23)):

$$\mu\|y^*_{\lambda^*} - y^*_{\tilde{\lambda}}\|^2 \leq \langle \nabla_y L_{\tilde{\lambda},\alpha}(x, y^*_{\tilde{\lambda}}) - \nabla_y L_{\lambda^*,\alpha}(x, y^*_{\tilde{\lambda}}), y^*_{\lambda^*} - y^*_{\tilde{\lambda}} \rangle$$
$$(\text{since } \nabla_y L_{\tilde{\lambda},\alpha}(x, y^*_{\tilde{\lambda}}) = 0)$$

Applying the Cauchy-Schwarz inequality:

$$\mu\|y^*_{\lambda^*} - y^*_{\tilde{\lambda}}\|^2 \leq \|\nabla_y L_{\tilde{\lambda},\alpha}(x, y^*_{\tilde{\lambda}}) - \nabla_y L_{\lambda^*,\alpha}(x, y^*_{\tilde{\lambda}})\| \cdot \|y^*_{\lambda^*} - y^*_{\tilde{\lambda}}\|$$

If $y^*_{\lambda^*} \neq y^*_{\tilde{\lambda}}$, we can divide by $\|y^*_{\lambda^*} - y^*_{\tilde{\lambda}}\|$:

$$\mu\|y^*_{\lambda^*} - y^*_{\tilde{\lambda}}\| \leq \|\nabla_y L_{\lambda^*,\alpha}(x, y^*_{\tilde{\lambda}}) - \nabla_y L_{\tilde{\lambda},\alpha}(x, y^*_{\tilde{\lambda}})\|$$

Lemma 4.1 states that for any fixed $(x,y)$, $\|\nabla L_{\lambda^*,\alpha}(x,y) - \nabla L_{\tilde{\lambda},\alpha}(x,y)\| \leq O(\alpha_1\delta + \alpha_2\delta)$. This implies there exists a constant, which we identify with $C_{\text{sol}}$ from the lemma statement (where $C_{\text{sol}}$ depends on $C_\lambda$ and $M_{\nabla h}$), such that:

$$\|\nabla L_{\lambda^*,\alpha}(x, y^*_{\tilde{\lambda}}) - \nabla L_{\tilde{\lambda},\alpha}(x, y^*_{\tilde{\lambda}})\| \leq C_{\text{sol}}(\alpha_1 + \alpha_2)\delta$$

Substituting this into the inequality above:

$$\mu\|y^*_{\lambda^*} - y^*_{\tilde{\lambda}}\| \leq C_{\text{sol}}(\alpha_1 + \alpha_2)\delta$$

Dividing by $\mu$ (which is positive as $\mu = \Omega(\alpha\mu_g)$ and $\mu_g > 0$, $\alpha > 0$) yields the result:

$$\|y^*_{\lambda^*,\alpha}(x) - y^*_{\tilde{\lambda},\alpha}(x)\| \leq \frac{C_{\text{sol}}}{\mu}(\alpha_1 + \alpha_2)\delta$$

If $y^*_{\lambda^*} = y^*_{\tilde{\lambda}}$, the inequality holds trivially. This completes the proof. $\square$

**Lemma 4.3** (Hypergradient Bias Bound). *Let $\nabla_x L_{\lambda,\alpha}(x,y)$ denote the partial gradient of the penalty Lagrangian with respect to $x$. Assume it is $L_{H,y}$-Lipschitz in $y$ and $L_{H,\lambda}$-Lipschitz in $\lambda$. With $\alpha_1 = \alpha^{-2}$, $\alpha_2 = \alpha^{-4}$, choose $\delta = \Theta(\alpha^3)$ and suppose $\|\tilde{y}(x) - y^*_{\tilde{\lambda},\alpha}(x)\| \le \delta$ and $\|\tilde{\lambda}(x) - \lambda^*(x)\| \le C_\lambda \delta$. If $L_{\lambda^*,\alpha}(x,\cdot)$ is $\mu$-strongly convex with $\mu \ge c_\mu \alpha^{-2}$, then*

$$\|\mathbb{E}[\nabla \tilde{F}(x)] - \nabla F(x)\| \le C_{bias}\alpha,$$

*where $C_{bias}$ depends only on $L_{H,y}$, $L_{H,\lambda}$, $C_g$, $C_\lambda$, $c_\mu$, and the penalty constant $C_{pen}$. Here $L_{H,y}$ and $L_{H,\lambda}$ are the Lipschitz constants of $\nabla_x L_{\lambda,\alpha}(x,y)$ with respect to $y$ and $\lambda$ (from the lemma assumptions); $C_g$ is the Lipschitz constant of $\nabla_y g$ (from Assumption 3.1(ii)); $C_\lambda$ is an upper bound on $\|\lambda^*(x)\|$ (guaranteed by strong convexity and LICQ); $c_\mu$ is a positive constant linking the lower-level strong convexity to $\alpha^{-2}$ (see Lemma 4.2); and $C_{pen}$ is the penalty parameter in our formulation (determined by the choice of $\alpha_1$ and $\alpha_2$ sufficiently large such that the penalty term dominates any curvature of $f$).*

*Proof.* The quantity to bound is the bias $\|\mathbb{E}[\nabla \tilde{F}(x)] - \nabla F(x)\| = \|\nabla_x L_{\tilde{\lambda},\alpha}(x,\tilde{y}(x)) - \nabla F(x)\|$. We decompose this error into three parts using the triangle inequality:

$$\|\nabla_x L_{\tilde{\lambda},\alpha}(x,\tilde{y}(x)) - \nabla F(x)\| \le \underbrace{\|\nabla_x L_{\tilde{\lambda},\alpha}(x,\tilde{y}(x)) - \nabla_x L_{\tilde{\lambda},\alpha}(x,y^*_{\tilde{\lambda},\alpha}(x))\|}_{T_1} \tag{24}$$

$$+ \underbrace{\|\nabla_x L_{\tilde{\lambda},\alpha}(x,y^*_{\tilde{\lambda},\alpha}(x)) - \nabla_x L_{\lambda^*,\alpha}(x,y^*_{\lambda^*,\alpha}(x))\|}_{T_2} \tag{25}$$

$$+ \underbrace{\|\nabla_x L_{\lambda^*,\alpha}(x,y^*_{\lambda^*,\alpha}(x)) - \nabla F(x)\|}_{T_3} \tag{26}$$

$(T_1)$: This term bounds the error from the inexact minimization of $L_{\tilde{\lambda},\alpha}(x,\cdot)$. Using the $L_{H,y}$-Lipschitz continuity of $\nabla_x L_{\tilde{\lambda},\alpha}(x,y)$ with respect to $y$ (as assumed in the lemma statement) and the condition $\|\tilde{y}(x) - y^*_{\tilde{\lambda},\alpha}(x)\| \le \delta$ (from the lemma statement, where $\delta = \Theta(\alpha^3)$):

$$T_1 \le L_{H,y}\|\tilde{y}(x) - y^*_{\tilde{\lambda},\alpha}(x)\| \le L_{H,y}\delta = O(\delta). \tag{27}$$

$(T_2)$: This term bounds the error from using the approximate dual $\tilde{\lambda}(x)$ instead of the true dual $\lambda^*(x)$ in defining the penalty Lagrangian and its minimizer. Using the triangle inequality:

$$T_2 \le \|\nabla_x L_{\tilde{\lambda},\alpha}(x,y^*_{\tilde{\lambda},\alpha}(x)) - \nabla_x L_{\tilde{\lambda},\alpha}(x,y^*_{\lambda^*,\alpha}(x))\|$$

$$+ \|\nabla_x L_{\tilde{\lambda},\alpha}(x,y^*_{\lambda^*,\alpha}(x)) - \nabla_x L_{\lambda^*,\alpha}(x,y^*_{\lambda^*,\alpha}(x))\|. \tag{28}$$

The first part of the sum is bounded by $L_{H,y}\|y^*_{\tilde{\lambda},\alpha}(x) - y^*_{\lambda^*,\alpha}(x)\|$ (using the assumed $L_{H,y}$-Lipschitz continuity of $\nabla_x L_{\tilde{\lambda},\alpha}(x,y)$ w.r.t $y$). The second part is bounded by $L_{H,\lambda}\|\tilde{\lambda}(x) - \lambda^*(x)\|$ (using the assumed $L_{H,\lambda}$-Lipschitz continuity of $\nabla_x L_{\cdot,\alpha}(x,y^*_{\lambda^*,\alpha}(x))$ w.r.t the dual variable). Invoking Lemma 4.2 for $\|y^*_{\tilde{\lambda},\alpha}(x) - y^*_{\lambda^*,\alpha}(x)\| \le \frac{C_{sol}}{\mu}(\alpha_1 + \alpha_2)\delta$, and using the condition $\|\tilde{\lambda}(x) - \lambda^*(x)\| \le C_\lambda \delta$ (from Assumption 3.3, with $\delta = \Theta(\alpha^3)$ as per this lemma's setup):

$$T_2 \le L_{H,y} \cdot \frac{C_{sol}}{\mu}(\alpha_1 + \alpha_2)\delta + L_{H,\lambda} \cdot C_\lambda \delta \tag{29}$$

$$= O\left(\frac{(\alpha_1 + \alpha_2)\delta}{\mu}\right) + O(\delta). \tag{30}$$

Given $\alpha_1 = \alpha^{-2}$, $\alpha_2 = \alpha^{-4}$, so $(\alpha_1 + \alpha_2) = O(\alpha^{-4})$. With $\delta = \Theta(\alpha^3)$ and $\mu = \Theta(\alpha^{-2})$ (from the lemma condition $\mu \ge c_\mu \alpha^{-2}$), the first term is $O\left(\frac{\alpha^{-4}\alpha^3}{\alpha^{-2}}\right) = O(\alpha)$. The second term $O(\delta)$ is $O(\alpha^3)$. Thus, $T_2 = O(\alpha)$.

$(T_3)$: This term measures the inherent approximation error of the idealized penalty method (using true $\lambda^*$ and exact minimization $y^*_{\lambda^*,\alpha}(x)$) with respect to the true hypergradient $\nabla F(x)$. As per the lemma's setup, this is bounded by:

$$T_3 = \|\nabla_x L_{\lambda^*,\alpha}(x,y^*_{\lambda^*,\alpha}(x)) - \nabla F(x)\| \le C_{pen}\alpha, \tag{31}$$

for some problem-dependent constant $C_{\text{pen}}$.

Combining Terms: Summing the bounds for $T_1, T_2$, and $T_3$, with $\delta = \Theta(\alpha^3)$:

$$\|\nabla_x L_{\tilde{\lambda}, \alpha}(x, \tilde{y}(x)) - \nabla F(x)\| \leq O(\delta) + O(\alpha) + O(C_{\text{pen}}\alpha) \tag{32}$$

$$= O(\alpha^3) + O(\alpha) + O(\alpha) = O(\alpha). \tag{33}$$

Since $\mathbb{E}[\nabla \tilde{F}(x)] = \nabla_x L_{\tilde{\lambda}, \alpha}(x, \tilde{y}(x))$, we conclude that $\|\mathbb{E}[\nabla \tilde{F}(x)] - \nabla F(x)\| \leq C_{\text{bias}}\alpha$. The conditions $\delta = \Theta(\alpha^3)$ and $\mu = \Theta(\alpha^{-2})$ ensure that all error components are either $O(\alpha)$ or of a smaller order. $\qquad\square$

**Lemma 4.4** (Variance Bound)**.** *Under Assumption 3.3 (i)–(ii), let $\sigma^2$ be a uniform bound on*

$\text{Var}_{x, \tilde{\lambda}, \tilde{y}}\big(\nabla_x \tilde{L}_{\tilde{\lambda}, \alpha}(x, \tilde{y}; \xi)\big).$

*With a mini-batch of $N_g$ i.i.d. samples in Algorithm 1, the conditional variance of the hypergradient estimate satisfies*

$$\text{Var}_{x, \tilde{\lambda}, \tilde{y}}\big(\nabla \tilde{F}(x)\big) \leq \frac{\sigma^2}{N_g}.$$

*Proof.* Let $G_j = \nabla_x \tilde{L}_{\tilde{\lambda}, \alpha}(x, \tilde{y}; \xi_j)$ for $j = 1, \ldots, N_g$. Conditional on $x, \tilde{\lambda}, \tilde{y}$, these $G_j$ are i.i.d. random vectors with mean $\mathbb{E}_{x, \tilde{\lambda}, \tilde{y}}[G_j] = \nabla_x L_{\tilde{\lambda}, \alpha}(x, \tilde{y}(x)) = \mathbb{E}_{x, \tilde{\lambda}, \tilde{y}}[\nabla \tilde{F}(x)].$

The conditional variance of the averaged estimator is:

$$\text{Var}_{x, \tilde{\lambda}, \tilde{y}}(\nabla \tilde{F}(x)) = \text{Var}_{x, \tilde{\lambda}, \tilde{y}}\left(\frac{1}{N_g} \sum_{j=1}^{N_g} G_j\right) = \frac{1}{N_g^2} \text{Var}_{x, \tilde{\lambda}, \tilde{y}}\left(\sum_{j=1}^{N_g} G_j\right) \tag{34}$$

Since the $G_j$ are independent conditional on $x, \tilde{\lambda}, \tilde{y}$, we have:

$$\text{Var}_{x, \tilde{\lambda}, \tilde{y}}\left(\sum_{j=1}^{N_g} G_j\right) = \sum_{j=1}^{N_g} \text{Var}_{x, \tilde{\lambda}, \tilde{y}}(G_j) \tag{35}$$

By our Assumption 3.3(ii), $\text{Var}_{x, \tilde{\lambda}, \tilde{y}}(G_j) \leq \sigma^2$ for all $j$. Therefore:

$$\text{Var}_{x, \tilde{\lambda}, \tilde{y}}(\nabla \tilde{F}(x)) = \frac{1}{N_g^2} \sum_{j=1}^{N_g} \text{Var}_{x, \tilde{\lambda}, \tilde{y}}(G_j) \leq \frac{1}{N_g^2} \sum_{j=1}^{N_g} \sigma^2 = \frac{N_g \sigma^2}{N_g^2} = \frac{\sigma^2}{N_g} \tag{36}$$

Thus, the variance of the hypergradient estimator is bounded by $\frac{\sigma^2}{N_g}$. $\qquad\square$

**Theorem 4.1** (Accuracy of Stochastic Hypergradient)**.** *Let $\nabla \tilde{F}(x)$ be the output of Algorithm 1 with penalty parameters $\alpha_1 = \alpha^{-2}, \alpha_2 = \alpha^{-4}$, and inner accuracy $\delta = O(\alpha^3)$. There exists a constant $C_{bias}$ such that:*

$$\mathbb{E}[\|\nabla \tilde{F}(x) - \nabla F(x)\|^2] \leq 2C_{bias}^2 \alpha^2 + \frac{2\sigma^2}{N_g}.$$

*Proof.* i) Using the bias-variance decomposition and properties of conditional expectation:

$$\mathbb{E}[\|\nabla \tilde{F}(x) - \nabla F(x)\|^2] = \mathbb{E}[\|\nabla \tilde{F}(x) - \mathbb{E}[\nabla \tilde{F}(x)] + \mathbb{E}[\nabla \tilde{F}(x)] - \nabla F(x)\|^2] \tag{37}$$

By the inequality $\|a + b\|^2 \leq 2\|a\|^2 + 2\|b\|^2$:

$$\mathbb{E}[\|\nabla \tilde{F}(x) - \nabla F(x)\|^2] \leq 2\mathbb{E}[\|\nabla \tilde{F}(x) - \mathbb{E}[\nabla \tilde{F}(x)]\|^2] + 2\|\mathbb{E}[\nabla \tilde{F}(x)] - \nabla F(x)\|^2 \tag{38}$$

The first term is the expected conditional variance:

$$\mathbb{E}[\|\nabla \tilde{F}(x) - \mathbb{E}[\nabla \tilde{F}(x)]\|^2] = \mathbb{E}[\mathbb{E}_{x,\tilde{\lambda},\tilde{y}}[\|\nabla \tilde{F}(x) - \mathbb{E}[\nabla \tilde{F}(x)]\|^2]] \tag{39}$$

$$= \mathbb{E}[\text{Var}_{x,\tilde{\lambda},\tilde{y}}(\nabla \tilde{F}(x))] \tag{40}$$

From Lemma 4.4, we know that $\text{Var}_{x,\tilde{\lambda},\tilde{y}}(\nabla \tilde{F}(x)) \leq \frac{\sigma^2}{N_g}$. Therefore:

$$\mathbb{E}[\|\nabla \tilde{F}(x) - \mathbb{E}[\nabla \tilde{F}(x)]\|^2] \leq \frac{\sigma^2}{N_g} \tag{41}$$

The second term is the squared bias, which from Lemma 4.3 is bounded by:

$$\|\mathbb{E}[\nabla \tilde{F}(x)] - \nabla F(x)\|^2 \leq (C_{\text{bias}}\alpha)^2 = C_{\text{bias}}^2\alpha^2 \tag{42}$$

Combining these bounds:

$$\mathbb{E}[\|\nabla \tilde{F}(x) - \nabla F(x)\|^2] \leq 2 \cdot \frac{\sigma^2}{N_g} + 2 \cdot C_{\text{bias}}^2\alpha^2 \tag{43}$$

$$= 2C_{\text{bias}}^2\alpha^2 + \frac{2\sigma^2}{N_g} \tag{44}$$

$\square$

**Lemma 4.5** (Inner-loop Oracle Complexity). *Fix $\alpha > 0$ and set $\alpha_1 = \alpha^{-2}$, $\alpha_2 = \alpha^{-4}$, $\delta = \Theta(\alpha^3)$. Let $g(x, \cdot)$ be $\mu_g$-strongly convex and $C_g$-smooth, and the stochastic oracles of Assumption 3.3 have variance $\sigma^2$. Choose the mini-batch size $N_g = \sigma^2/\alpha^2$. Running Algorithm 1 with $\tilde{O}(\alpha^{-2})$ stochastic first-order oracle (SFO) calls in its inner loops yields a stochastic inexact gradient $\nabla \tilde{F}(x)$ characterized by bias of $O(\alpha)$ and variance of $O(\alpha^2)$.*

*Proof.* We count the stochastic-gradient oracle calls made in one execution of Algorithm 1. The inner tolerance is $\delta = \Theta(\alpha^3)$.

C1. Lower-level pair $(\tilde{y}^*, \tilde{\lambda}^*)$: For every outer iterate $x$, the constrained LL objective $g(x, \cdot)$ is $\mu_g$-strongly convex and $C_g$-smooth (Assumption 3.1). A stochastic primal-dual (SPD) algorithm with mini-batches satisfies linear convergence $\mathbb{E}\|y_t - y^*\|^2 \leq \left(1 - \frac{1}{\kappa_g}\right)^t D_0^2$, $\quad \kappa_g := C_g/\mu_g$ Hence

$$t_1 = O(\kappa_g \log(1/\delta)) = O\left(\frac{C_g}{\mu_g} \log \frac{1}{\delta}\right)$$

oracle calls give $\|\tilde{y}^* - y^*\|$, $\|\tilde{\lambda}^* - \lambda^*\| \leq \delta$.

C2. Penalty minimisation $(\tilde{y})$: With $\alpha_1 = \alpha^{-2}$ and $\alpha_2 = \alpha^{-4}$ we analyze $L_{\tilde{\lambda}^*,\alpha}(x, \cdot)$:

- *Strong convexity.* The term $\alpha_1 g$ contributes $\alpha_1 \mu_g$; the smooth term $f$ can subtract at most $C_f$ curvature. For sufficiently small $\alpha$, $\mu_{\text{pen}} \geq \alpha_1 \mu_g/2$.

- *Smoothness.* Because each $h_i$ is affine in $y$, the quadratic penalty has Hessian bounded by $\alpha_2 \|B\|^2$, so $L_{\text{pen}} = \Theta(\alpha_2)$

Therefore the condition number is

$$\kappa_{\text{pen}} = \frac{L_{\text{pen}}}{\mu_{\text{pen}}} = \Theta(\alpha^{-2}/\mu_g).$$

A linear-rate variance-reduced method (SVRG) requires $t_2 = O(\kappa_{\text{pen}} \log(1/\delta)) = O(\alpha^{-2} \log(1/\delta)/\mu_g)$ oracle calls to attain $\|\tilde{y} - y^*_{\tilde{\lambda}^*,\alpha}\| \leq \delta$Johnson and Zhang (2013).

C3. Total inner cost: Summing $t_1$ and $t_2$ and adding the mini-batch evaluations:

$$\text{cost}(x) = O\Big(\big(\tfrac{C_g}{\mu_g} + \tfrac{\alpha^{-2}}{\mu_g}\big)\log\tfrac{1}{\delta}\Big) \;+\; N_g.$$

Because $\delta = \Theta(\alpha^3)$, $\log(1/\delta) = 3\log(1/\alpha)$ (absorbed into $\tilde{O}(\cdot)$) and $\alpha^{-2}$ dominates $C_g$ for small $\alpha$, so

$$\text{cost}(x) = \tilde{O}\big(\alpha^{-2}/\mu_g\big) + N_g.$$

Using Lemma 4.4, $N_g$ should satisfy $\sigma/\sqrt{N_g} \;\asymp\; \alpha$, hence $N_g = \Theta(\sigma^2/\alpha^2)$. Plugging in, $\text{cost}(x) = \tilde{O}(\alpha^{-2})$ ( constants depending on $\mu_g$ and $\sigma^2$ are absorbed).

With this batch size, $\mathbb{E}\big\|\tilde{\nabla}F(x) - \nabla F(x)\big\| \leq O(\alpha) + \sigma/\sqrt{N_g} = O(\alpha)$, so the oracle outputs an $\alpha$-accurate hyper-gradient.

Set $\alpha = \Theta(\varepsilon)$ for outer-loop tolerance $\varepsilon$; the inner cost becomes $\tilde{O}(\varepsilon^{-2})$ □

**Theorem 5.1** (Convergence with Stochastic Hypergradient Oracle). *Suppose $F : \mathbb{R}^n \to \mathbb{R}$ is $L_F$-Lipschitz. Let $\nabla\tilde{F}(\cdot)$ be a stochastic hypergradient oracle satisfying:*

1. *Bias bound: $\|\mathbb{E}[\nabla\tilde{F}(x)] - \nabla F(x)\| \leq C_{bias}\alpha$*

2. *Variance bound: $\mathbb{E}[\|\nabla\tilde{F}(x) - \mathbb{E}[\nabla\tilde{F}(x)]\|^2] \leq \frac{\sigma^2}{N_g}$*

*Then running Algorithm 2 with parameters $D = \Theta(\frac{\delta\epsilon^2}{L_F^2})$, $\eta = \Theta(\frac{\delta\epsilon^3}{L_F^4})$, and $N_g = \Theta(\frac{\sigma^2}{\alpha^2})$ outputs a point $x_{out}$ such that $\mathbb{E}[dist(\mathbf{0}, \partial_\delta F(x_{out}))] \leq \epsilon + O(\alpha)$, using $T = O(\frac{(F(x_0)-\inf F)L_F^2}{\delta\epsilon^3})$ calls to $\nabla\tilde{F}(\cdot)$.*

*Proof.* For any $t \in [T]$, since $x_t = x_{t-1} + \Delta_t$, we have by the fundamental theorem of calculus:

$$F(x_t) - F(x_{t-1}) = \int_0^1 \langle \nabla F(x_{t-1} + s\Delta_t), \Delta_t \rangle ds \tag{45}$$

$$= \mathbb{E}_{s_t \sim \text{Unif}[0,1]}[\langle \nabla F(x_{t-1} + s_t\Delta_t), \Delta_t \rangle] \tag{46}$$

$$= \mathbb{E}[\langle \nabla F(z_t), \Delta_t \rangle] \tag{47}$$

where equation (47) follows from our algorithm's definition of $z_t = x_{t-1} + s_t\Delta_t$. Summing over $t \in [T] = [K \times M]$:

$$\inf F \leq F(x_T) = F(x_0) + \sum_{t=1}^{T} \mathbb{E}[\langle \nabla F(z_t), \Delta_t \rangle] \tag{48}$$

$$= F(x_0) + \underbrace{\sum_{k=1}^{K}\sum_{m=1}^{M} \mathbb{E}[\langle \nabla F(z_{(k-1)M+m}), \Delta_{(k-1)M+m} - u_k \rangle]}_{\text{regret of online gradient descent}}$$

$$+ \underbrace{\sum_{k=1}^{K}\sum_{m=1}^{M} \mathbb{E}[\langle \nabla F(z_{(k-1)M+m}), u_k \rangle]}_{\text{Gradient norm}} \tag{49}$$

where we've added and subtracted $\langle \nabla F(z_t), u_k \rangle$ in (49) for any sequence of reference points $u_1, \ldots, u_K \in \mathbb{R}^d$ satisfying $\|u_i\| \leq D$ for all $i$.

The first double sum represents the regret of online gradient descent with stochastic gradients. For any $t \in [T]$:

$$\|\Delta_{t+1} - u_k\|^2 = \|\text{clip}_D(\Delta_t - \eta\tilde{g}_t) - u_k\|^2 \tag{50}$$

$$\leq \|\Delta_t - \eta\tilde{g}_t - u_k\|^2 \tag{51}$$

$$= \|\Delta_t - u_k\|^2 + \eta^2\|\tilde{g}_t\|^2 - 2\eta\langle \Delta_t - u_k, \tilde{g}_t \rangle \tag{52}$$

where (51) follows since projection onto a convex set decreases distance. Rearranging (52):

$$\langle \tilde{g}_t, \Delta_t - u_k \rangle \leq \frac{\|\Delta_t - u_k\|^2 - \|\Delta_{t+1} - u_k\|^2}{2\eta} + \frac{\eta \|\tilde{g}_t\|^2}{2} \tag{53}$$

Now, we decompose the key inner product using the bias-variance structure of our stochastic gradient oracle:

$$\mathbb{E}[\langle \nabla F(z_t), \Delta_t - u_k \rangle] = \mathbb{E}[\langle \tilde{g}_t, \Delta_t - u_k \rangle] + \mathbb{E}[\langle \nabla F(z_t) - \tilde{g}_t, \Delta_t - u_k \rangle] \tag{54}$$

**First term in eq. (54):** For the first term in (54), using inequality (53):

$$\mathbb{E}[\langle \tilde{g}_t, \Delta_t - u_k \rangle] \leq \mathbb{E}\left[\frac{\|\Delta_t - u_k\|^2 - \|\Delta_{t+1} - u_k\|^2}{2\eta} + \frac{\eta \|\tilde{g}_t\|^2}{2}\right] \tag{55}$$

For the expected squared norm in (55), using the bias-variance decomposition and the $L$-Lipschitz property of $F$:

$$\mathbb{E}[\|\tilde{g}_t\|^2] = \mathbb{E}[\|\mathbb{E}_{z_t}[\tilde{g}_t] + (\tilde{g}_t - \mathbb{E}_{z_t}[\tilde{g}_t])\|^2] \tag{56}$$

$$\leq \mathbb{E}[\|\mathbb{E}_{z_t}[\tilde{g}_t]\|^2] + \mathbb{E}[\|\tilde{g}_t - \mathbb{E}_{z_t}[\tilde{g}_t]\|^2] \tag{57}$$

$$\leq L^2 + \frac{\sigma^2}{N_g} \tag{58}$$

$$= L^2 + O(\alpha^2) = O(1) \quad \text{(for small } \alpha = o(1) \text{ and } L \text{ is a given constant)} \tag{59}$$

where (57) follows from the orthogonality of bias and variance terms. Therefore, we have:

$$\mathbb{E}[\langle \tilde{g}_t, \Delta_t - u_k \rangle] \leq \mathbb{E}\left[\frac{\|\Delta_t - u_k\|^2 - \|\Delta_{t+1} - u_k\|^2}{2\eta}\right] + O(\eta) \tag{60}$$

**Second term in eq. (54):** based on Cauchy-Schwarz inequality and noting that both $\|\Delta_t\| \leq D$ and $\|u_k\| \leq D$ by construction, we have:

$$\mathbb{E}[\langle \nabla F(z_t) - \tilde{g}_t, \Delta_t - u_k \rangle] \leq \mathbb{E}[\|\nabla F(z_t) - \tilde{g}_t\| \cdot \|\Delta_t - u_k\|] \tag{61}$$

$$\leq 2D \cdot \mathbb{E}[\|\nabla F(z_t) - \tilde{g}_t\|] \tag{62}$$

By triangle inequality and the properties of our stochastic oracle:

$$\mathbb{E}[\|\nabla F(z_t) - \tilde{g}_t\|] \leq \mathbb{E}[\|\nabla F(z_t) - \mathbb{E}_{z_t}[\tilde{g}_t]\|] + \mathbb{E}[\|\mathbb{E}_{z_t}[\tilde{g}_t] - \tilde{g}_t\|] \tag{63}$$

$$\leq C_{\text{bias}}\alpha + \mathbb{E}[\|\mathbb{E}_{z_t}[\tilde{g}_t] - \tilde{g}_t\|] \tag{64}$$

For the variance term in (64), by Jensen's inequality:

$$\mathbb{E}[\|\mathbb{E}_{z_t}[\tilde{g}_t] - \tilde{g}_t\|] \leq \sqrt{\mathbb{E}[\|\mathbb{E}_{z_t}[\tilde{g}_t] - \tilde{g}_t\|^2]} \tag{65}$$

$$= \sqrt{\mathbb{E}[\mathbb{E}_{z_t}[\|\mathbb{E}_{z_t}[\tilde{g}_t] - \tilde{g}_t\|^2]]} \tag{66}$$

$$= \sqrt{\mathbb{E}[\text{Var}_{z_t}(\tilde{g}_t)]} \leq \frac{\sigma}{\sqrt{N_g}} \tag{67}$$

where (66) follows from the tower property of conditional expectation, and (67) uses our variance bound assumption. When $N_g = \Theta(\frac{\sigma^2}{\alpha^2})$ ensures:

$$\mathbb{E}[\|\nabla F(z_t) - \tilde{g}_t\|] \leq C_{\text{bias}}\alpha + \frac{\sigma}{\sqrt{N_g}} = C_{\text{bias}}\alpha + O(\alpha) = O(\alpha) \tag{68}$$

Therefore, combining (62) and (68), we can bound the second term in eq. (54) by:

$$\mathbb{E}[\langle \nabla F(z_t) - \tilde{g}_t, \Delta_t - u_k \rangle] \leq 2D \cdot O(\alpha) = O(D\alpha) \tag{69}$$

**Putting together first term and second term** : Combining (60) and (69):

$$\mathbb{E}[\langle \nabla F(z_t), \Delta_t - u_k \rangle] \le \mathbb{E}\left[\frac{\|\Delta_t - u_k\|^2 - \|\Delta_{t+1} - u_k\|^2}{2\eta}\right] + O(\eta) + O(D\alpha) \tag{70}$$

Summing (70) over $t = (k-1)M + m$ with $m = 1, \ldots, M$ for a fixed $k$:

$$\sum_{m=1}^{M} \mathbb{E}[\langle \nabla F(z_{(k-1)M+m}), \Delta_{(k-1)M+m} - u_k \rangle]$$

$$\le \sum_{m=1}^{M} \mathbb{E}\left[\frac{\|\Delta_{(k-1)M+m} - u_k\|^2 - \|\Delta_{(k-1)M+m+1} - u_k\|^2}{2\eta}\right] + \sum_{m=1}^{M} O(\eta) + \sum_{m=1}^{M} O(D\alpha) \tag{71}$$

$$\le \frac{\mathbb{E}[\|\Delta_{(k-1)M+1} - u_k\|^2 - \|\Delta_{(k-1)M+M+1} - u_k\|^2]}{2\eta} + O(M\eta) + O(MD\alpha) \tag{72}$$

Since $\|\Delta_t\| \le D$ and $\|u_k\| \le D$, we have $\|\Delta_t - u_k\| \le 2D \; \forall t$. Therefore, we can further bound (72) by:

$$\frac{\mathbb{E}[\|\Delta_{(k-1)M+1} - u_k\|^2 - \|\Delta_{(k-1)M+M+1} - u_k\|^2]}{2\eta} + O(M\eta) + O(MD\alpha) \tag{73}$$

$$\le \frac{4D^2}{2\eta} + O(M\eta) + O(MD\alpha)$$

$$= O\left(\frac{D^2}{\eta} + M\eta + MD\alpha\right) \tag{74}$$

Since this inequality holds for all $\eta \in \mathbb{R}_+$, we can choose $\eta = O\left(\frac{D}{\sqrt{M}}\right)$ to minimize the upper bound to get the tightest upper bound:

$$\sum_{m=1}^{M} \mathbb{E}[\langle \nabla F(z_{(k-1)M+m}), \Delta_{(k-1)M+m} - u_k \rangle] \le O(D\sqrt{M} + MD\alpha) \tag{75}$$

PART 2: BOUNDING THE REGRET OF ONLINE GRADIENT DESCENT IN EQ. (49)

For the second term in (49), we choose $u_k$ strategically to extract the Goldstein subdifferential:

$$u_k = -D \cdot \frac{\sum_{m=1}^{M} \nabla F(z_{(k-1)M+m})}{\|\sum_{m=1}^{M} \nabla F(z_{(k-1)M+m})\|} \tag{76}$$

With this choice of $u_k$:

$$\sum_{m=1}^{M} \langle \nabla F(z_{(k-1)M+m}), u_k \rangle = -D \cdot \left\|\sum_{m=1}^{M} \nabla F(z_{(k-1)M+m})\right\| \tag{77}$$

$$= -DM \cdot \left\|\frac{1}{M} \sum_{m=1}^{M} \nabla F(z_{(k-1)M+m})\right\| \tag{78}$$

Substituting (75) and (78) into (49), and then into (48):

$$F(x_0) - \inf F \ge \sum_{k=1}^{K} \left[-O(D\sqrt{M}) - O(MD\alpha) + DM \cdot \left\|\frac{1}{M} \sum_{m=1}^{M} \nabla F(z_{(k-1)M+m})\right\|\right] \tag{79}$$

Solving for the average over $k$:

$$\frac{1}{K} \sum_{k=1}^{K} \left\|\frac{1}{M} \sum_{m=1}^{M} \nabla F(z_{(k-1)M+m})\right\| \le \frac{F(x_0) - \inf F}{DMK} + O\left(\frac{1}{\sqrt{M}}\right) + O(\alpha) \tag{80}$$

For the randomly chosen output $x_{\text{out}} \sim \text{Uniform}\{x_1, \ldots, x_K\}$:

$$\mathbb{E}\left[\left\|\frac{1}{M}\sum_{m=1}^{M}\nabla F(z_{(k-1)M+m})\right\|\right] \leq \frac{F(x_0) - \inf F}{DMK} + O(\frac{1}{\sqrt{M}}) + O(\alpha) \tag{81}$$

The key insight is that these averages approximate the Goldstein subdifferential. Since $\|z_{(k-1)M+m} - x_k\| \leq MD \leq \delta$ (by our choice of $M = \lfloor\frac{\delta}{D}\rfloor$), we have:

$$\nabla F(z_{(k-1)M+m}) \in \partial_\delta F(x_k) \text{ for all } m \in [M] \tag{82}$$

By convexity of the Goldstein subdifferential:

$$\frac{1}{M}\sum_{m=1}^{M}\nabla F(z_{(k-1)M+m}) \in \partial_\delta F(x_k) \tag{83}$$

Therefore, from (81) and (83):

$$\mathbb{E}[\text{dist}(0, \partial_\delta F(x_{\text{out}}))] \leq \frac{F(x_0) - \inf F}{DMK} + \Theta\left(\frac{1}{\sqrt{M}}\right) + \Theta(\alpha) \tag{84}$$

To achieve $\mathbb{E}[\text{dist}(0, \partial_\delta F(x_{\text{out}}))] = \Theta(\epsilon)$, we set $\alpha = \Theta(\epsilon)$ and balance the remaining terms:

$$\frac{F(x_0) - \inf F}{DMK} + \Theta\left(\frac{1}{\sqrt{M}}\right) \leq \epsilon \tag{85}$$

Let $C_0 = F(x_0) - \inf F$. We need both terms to be $\Theta(\epsilon)$:

$$\frac{C_0}{DMK} = \Theta(\epsilon) \tag{86}$$

$$\frac{1}{\sqrt{M}} = \Theta(\epsilon) \tag{87}$$

From (87), we get:

$$\frac{1}{\sqrt{M}} = \Theta(\epsilon) \implies M = \Theta\left(\frac{1}{\epsilon^2}\right) \tag{88}$$

Since $M = \lfloor\frac{\delta}{D}\rfloor$, we have $M \approx \frac{\delta}{D}$, which gives us:

$$\frac{\delta}{D} = \Theta\left(\frac{1}{\epsilon^2}\right) \implies D = \Theta\left(\delta\epsilon^2\right) \tag{89}$$

Let's set $D = \Theta(\delta\epsilon^2)$ and $M = \Theta\left(\frac{1}{\epsilon^2}\right)$ to satisfy this constraint. From (86), we can determine $K$:

$$\frac{C_0}{DMK} = \Theta(\epsilon) \implies K = \Theta\left(\frac{C_0}{DM\epsilon}\right) \tag{90}$$

Substituting our choices for $D$ and $M$:

$$K = \Theta\left(\frac{C_0}{\delta\epsilon^2 \cdot \frac{1}{\epsilon^2} \cdot \epsilon}\right) \tag{91}$$

$$= \Theta\left(\frac{C_0}{\delta\epsilon}\right) \tag{92}$$

Let's set $K = \Theta\left(\frac{C_0}{\delta\epsilon}\right)$ to satisfy this constraint. For the step size $\eta$, we need to ensure stability of the algorithm. Based on standard analysis of stochastic gradient methods, we typically set:

$$\eta = \Theta\left(\frac{D}{\sqrt{M}}\right) = \Theta\left(\delta\epsilon^2 \cdot \epsilon\right) = \Theta\left(\delta\epsilon^3\right) \tag{93}$$

Therefore, our final parameter settings are:

$$D = \Theta(\delta\epsilon^2) \tag{94}$$

$$M = \Theta\left(\frac{1}{\epsilon^2}\right) \tag{95}$$

$$K = \Theta\left(\frac{C_0}{\delta\epsilon}\right) \tag{96}$$

$$\eta = \Theta(\delta\epsilon^3) \tag{97}$$

Therefore, these parameter choices lead to $\mathbb{E}[\mathrm{dist}(0, \partial_\delta F(x_{\mathrm{out}}))] \leq \epsilon + O(\alpha)$

$\square$

