# OpenReview forum: "Bridging Constraints and Stochasticity: A Fully First-Order Method for Stochastic Bilevel Optimization with Linear Constraints"
_ICLR.cc/2026/Conference — Submitted to ICLR 2026_

### Official Review · Reviewer_WVrC · 2025-10-29

**Soundness:** 2
**Presentation:** 2
**Contribution:** 2
**Rating:** 2
**Confidence:** 4

**Summary:**

This paper establishes a theoretical milestone by bridging the gap between linear constrained and stochastic bilevel optimization, delivering a purely first-order, provably convergent algorithm. It lays the groundwork for extending efficient bilevel solvers to more realistic, noisy, and constrained ML applications like meta-learning, RL, and data reweighting.

**Strengths:**

1. This paper provides a finite-time stochastic convergence with linear constraints and first-order access.

2. This paper also has strong theoretical grounding (bias/variance analysis and Goldstein stationarity), and the proposed method has superior scalability for high-dimensional problems.

**Weaknesses:**

1. This paper announces that it provides the first finite-time convergence guarantees. However, there are several works about constraints in bilevel optimization, such as Overcoming Lower-Level Constraints in Bilevel Optimization: A Novel Approach with Regularized Gap Functions. Can the author provide some comparison?

2. It looks like the Assumption 3.1 (ii) asks lower-level $g$ to be strongly convex and also have a bounded gradient. Can the author verify this assumption?

3. This paper also employed additional assumptions compared to other bi-level works. Such as Assumption 3.1 (iii) and Assumption 3.2.

4. The paper is not well organized and is hard to read. Such as $\lambda^*(x)$ in line 151 is used before defined.

**Questions:**

1. Why is Assumption 3.2 necessary? In traditional bilevel optimization, this condition typically appears as a lemma rather than an assumption. Could the authors clarify what specific difficulty prevents deriving a similar lemma in the constrained setting?

2. Please compare the role and strength of this assumption with those used in other bilevel optimization works, particularly in constrained bilevel formulations.

3. Does the proposed problem have any practical applications? The current experiments appear overly simplified and resemble toy examples, which raises concerns about the real-world relevance of the proposed method.

---

> ### Author Response · Authors · 2025-11-26
>
> We thank your detailed feedback and updated the paper accordingly in the newest version. Below we respond point-by-point to each concern.
>
> > **1. This paper announces that it provides the first finite-time convergence guarantees. However, there are several works about constraints in bilevel optimization, such as Overcoming Lower-Level Constraints in Bilevel Optimization: A Novel Approach with Regularized Gap Functions. Can the author provide some comparison?**
>
> We appreciate this pointer. We have added explicit discussion of Yao et al. (2024) in Section 2. We now write: "Yao et al. (2024) propose a single-level reformulation using a doubly regularized gap function to handle general lower-level constraints, achieving a favorable $O(\epsilon^{-2})$ rate to an $\epsilon$-KKT stationary point under convexity assumptions and access to projection oracles. This approach avoids requiring strong convexity of $g$ or LICQ, but applies mainly to convex lower-level problems. In contrast, our method handles nonconvex upper-level objectives and leverages strong convexity of $g$; we focus on the Goldstein criterion rather than $\epsilon$-KKT. We clarify that their method is single-loop and Hessian-free (like ours) but requires the lower level to be convex, while we allow nonconvex $F(x)$ but assume strong convexity in $y$. The Conclusion notes that extending our method to nonlinear constraints (which the gap function method can handle) is a future direction.
>
> > **2. It looks like the Assumption 3.1 (ii) asks lower-level $g$ to be strongly convex and also have a bounded gradient. Can the author verify this assumption?**
>
> We have verified these conditions are internally consistent and clarified them. The bounded gradient assumption $|\nabla g(x,y)| \le L_g$ is standard in stochastic optimization to ensure the variance of stochastic gradients is bounded. In practice, this can be enforced by assuming compact domains or that the data distribution yields bounded gradients. The strong convexity $\mu_g > 0$ ensures a unique minimizer $y^*(x)$ for each $x$, necessary for our hypergradient formulation. We have added explanations right after Assumption 3.1 that this strong convexity ensures a unique lower-level minimizer $y^{\*}(x)$ for each $x$, which is necessary for our hypergradient formulation. The bounded gradient assumptions are standard and used to control variance and higher-order terms." We also added a reference to Facchinei & Pang (2003) noting that Lipschitz continuity of $y^{\*}(x)$ typically follows from strong convexity under constraint qualifications.
>
> > **3. Why is Assumption 3.2 necessary? In traditional bilevel optimization, this condition typically appears as a lemma rather than an assumption.**
>
> We have clarified this assumption and its role. In the paper, we now explicitly cite Facchinei & Pang (2003) after stating Assumption 3.2, noting that under strong convexity and LICQ, one can expect the solution map to be Lipschitz continuous in a neighborhood of each point. We assume global Lipschitzness for simplicity of analysis. We added: "Given the global Lipschitzness, the resulting UL objective $F(x)$ is then $L_F$-Lipschitz continuous (with $L_F \le L_{f,x} + L_{f,y}L_y$). This assumption guarantees that $y^*(x)$ varies Lipschitzly with $x$, which we use to control the error when linearizing $F(x_t)$ around $F(x_{t-1})$ during our convergence analysis." While it might be derivable from more primitive conditions (strong convexity + LICQ + bounded $\nabla^2 g$ yields local Lipschitz continuity by implicit function theorem; global Lipschitz might additionally need convex $X$ and linear constraints), we assume it to avoid a digression. The assumption's role is now clearly stated and justified.
>
> > **4. Does the proposed problem have any practical applications? The current experiments appear overly simplified.**
>
> We understand this concern. Practical applications of this method are meta-learning (hyperparameter optimization), constrained reinforcement learning, and large-scale empirical risk minimization with constraints. We emphasize that F2CSA is fully first-order and avoids Hessians, which is a significant practical advantage for large models. We highlight that F2CSA's scalability makes it suitable for high-dimensional problems with limited resources (as seen in our experiments, where it achieved approximately $3\times$ speedup over Hessian methods at $d=4000$). We acknowledge that our current experiments are limited but explain the key trade-offs observed (F2CSA is faster but the final loss is slightly higher than DSBLO's), which gives insight into when one might prefer our method in practice. We also state that we plan to include at least one experiment on a more realistic benchmark in the final version.

---

### Official Review · Reviewer_7guu · 2025-10-30

**Soundness:** 3
**Presentation:** 3
**Contribution:** 3
**Rating:** 6
**Confidence:** 3

**Summary:**

This paper studies stochastic bilevel optimization with linearly constrained lower-level (LL) problems, a setting where no prior work provides finite-time guarantees. The authors propose F2CSA (Fully First-order Constrained Stochastic Approximation)—a fully first-order method requiring only noisy gradients from the upper- and lower-level objectives. The key idea is to construct a stochastic inexact hypergradient oracle via a smoothed Lagrangian/penalty formulation with scaling parameters $\alpha_1 = \alpha^{-2}$ and $\alpha_2 = \alpha^{-4}$, together with inexact primal–dual LL solves.
They prove that the oracle has bias $O(\alpha)$ and variance $O(1/N_g)$, and that when used in a clipped nonsmooth outer loop, the algorithm converges to a $(\delta,\epsilon)$-Goldstein stationary point with total complexity $\tilde{O}(\delta^{-1}\epsilon^{-5})$—the first finite-time result for this class.

**Strengths:**

1. First finite-time guarantee for stochastic bilevel problems with linearly constrained LL subproblems, using a fully first-order method.
2. The presentation is clear.

**Weaknesses:**

1. The experimental evaluation is limited; additional large-scale experiments would be valuable to demonstrate the method’s scalability and practical relevance.
2. The LICQ assumption appears somewhat strong. Could the authors consider relaxing it to a weaker constraint qualification, or provide more discussion on why this assumption is essential for the current analysis?

**Questions:**

1. What other stationarity notions (beyond Goldstein stationary points) have been adopted in prior literature? A more comprehensive literature review on alternative stationarity metrics would strengthen the paper’s context.
2. I wonder whether variance-reduction or momentum techniques could further improve the theoretical complexity bounds within the proposed framework.

---

> ### Author Response · Authors · 2025-11-26
>
> We thank your detailed feedback and updated the paper accordingly in the newest version. Below we respond point-by-point to each concern.
>
> > **1. The experimental evaluation is limited; additional large-scale experiments would be valuable.**
>
> We acknowledge this limitation. Section 6 focuses on controlled synthetic problems to validate the theory. The Conclusion now explicitly states that we will add a real-world constrained bilevel task (e.g., bilevel adversarial problems with attack constraints) and additional metrics (constraint satisfaction, stationarity measures) in the final version. We also emphasize that F2CSA is applicable to high-dimensional constrained bilevel tasks in meta-learning, reinforcement learning, and large-scale hyperparameter tuning, and discuss the trade-off observed (speed vs. final loss) to clarify when practitioners might prefer our method.
>
> > **2. The LICQ assumption appears somewhat strong. Could the authors consider relaxing it?**
>
> We agree LICQ is strong. We have added a note after Assumption 3.1 that LICQ can potentially be relaxed to a weaker qualification (e.g., requiring a Slater condition or Mangasarian-Fromovitz conditions) at the cost of more complex analysis; we impose LICQ for theoretical tractability and to ensure differentiability of $y^{\*}(x)$. As future work, we will extend on structure-aware penalties that remain stable under weaker qualifications or partial degeneracy. If LICQ fails, the lower-level solution may not be differentiable in $x$ (multiple $\lambda^*$ may exist), complicating the hypergradient. Our analysis relies on differentiability via implicit function theory; relaxing LICQ would require subdifferential-based tools.
>
> > **3. What other stationarity notions (beyond Goldstein stationary points) have been adopted in prior literature?**
>
> We have added a discussion in Section 2 contrasting Goldstein's $(\delta,\epsilon)$-stationarity with $\epsilon$-KKT stationarity. We explain that Goldstein's notion is weaker and more suited for our nonsmooth setting, as it requires that no feasible perturbation within a $\delta$-radius can reduce $F(x)$ by more than $\epsilon$. In contrast, an $\epsilon$-KKT stationary point demands approximate satisfaction of all KKT conditions, which is stronger and typically requires smoother behavior or projection oracles. We cite Kornowski et al. (2024) as using Goldstein, and Lu & Mei (2024) and Yao et al. (2024) as using $\epsilon$-KKT. Before Theorem 5.1, we remind the reader that $\text{dist}(0,\partial_\delta F(x))\le \epsilon$ corresponds to $(\delta,\epsilon)$-Goldstein stationarity.
>
> > **4. I wonder whether variance-reduction or momentum techniques could further improve the theoretical complexity bounds.**
>
> We agree this is promising. The Introduction now states the result of $\tilde{O}(\delta^{-1}\epsilon^{-5})$ is one factor of $\epsilon$ away from the optimal stochastic rate, indicating potential room for improvement. As future work, we plan on using variance-reduced estimators or momentum (clipping-plus-momentum) to approach the optimal $\tilde O(\delta^{-1}\epsilon^{-4})$ dependence. We cite Yang et al. (2023) and related works. Technically, applying momentum is nontrivial due to the two-level structure and non-smoothness, but we suspect STORM-type variance reduction on the outer updates could reduce $\epsilon^{-5}$ to $\epsilon^{-4}$.

---

### Official Review · Reviewer_m2gH · 2025-10-31

**Soundness:** 2
**Presentation:** 1
**Contribution:** 2
**Rating:** 2
**Confidence:** 4

**Summary:**

This paper studies the linearly constrained stochastic bilevel optimization problem with first-order methods. The authors propose the algorithm F2CSA and provide convergence analysis.

**Strengths:**

1. The first-order methods for the constraint bilevel optimization problem have not been fully studied before.
2. The authors provide proof sketches for better understanding.

**Weaknesses:**

**There are lots of presentation problems that I doubt the correctness of the proof.**
1. In line 191, there is an incomplete sentence.
2. There is no explanation of Algorithm 1 before Remark 4.1. Therefore, there are a lot of undefined notations in it.
3. There is no update rule for $\tilde{\lambda}(x)$.
4. For the stochastic algorithm, are the authors sure that we can get $\\|\\tilde{y}^\ast(x)-y^*(x)\\|\\leq\mathcal{O}(\delta)$ rather than  in the expectation form with samples ($\mathbb{E}[\\|...\\|]\leq...$ with some samples $\xi$)? The same problem exists for $\lambda$.
5. In line 5 of Algorithm 1, what does "$\\|\leq\delta$" mean?
6. In line 246, $\alpha\geq\frac{2C_f}{\mu}$. This means that $\alpha$ is at a constant order. However, later, $\alpha$ is set to the $\epsilon$ order so that the algorithm converges. This contradiction makes me highly doubt the correctness of the proof.
7. In Lemma 4.3, $L_{H,y}$ and $L_{H,\lambda}$ are not defined, and the formulation is not given either.
8. For this linear constraint BO problem, I do not see how $h(x,y)$ impacts the convergence.

There are more problems that I have not listed yet. At this point, I believe the paper is far from ready.

**Questions:**

Please check the weakness part.

---

> ### Author Response · Authors · 2025-11-26
>
> We thank your detailed feedback and updated the paper accordingly in the newest version. Below we respond point-by-point to each concern.
>
> > **1. In line 191, there is an incomplete sentence.**
>
> We have added the missing definition in Section 1 where $S(x)$ denotes the feasible set of the lower-level problem (e.g., $S(x)=\mathbb{R}^m$ in the unconstrained case, or $S(x)=\lbrace y \colon h(x,y) \le 0 \rbrace$ for constrained cases).
>
> > **2. There is no explanation of Algorithm 1 before Remark 4.1, leading to undefined notations.**
>
> We have clarified all notations in Algorithm 1. The input now explicitly refers to the variance bound $\sigma^2$ (from Assumption 3.3). Step 3 clarifies that $(\tilde{y}^{\ast}(x), \tilde{\lambda}(x))$ are approximate lower-level primal-dual solutions obtained via a stochastic primal-dual method. We define $y^{\ast}(x)$ and $\lambda^{\ast}(x)$ earlier in the paper as the true lower-level minimizer and optimal multiplier. A notation paragraph at the start of Section 4 explicitly lists all definitions. Step 5 now computes $\tilde{y}(x)$ by stochastic gradient steps such that $\lVert \tilde{y}(x) - y^{\ast}{\tilde{\lambda},\alpha}(x) \rVert \le \delta$, where $y^{\ast}{\tilde{\lambda},\alpha}(x) := \arg\min_{y} L_{\tilde{\lambda},\alpha}(x, y)$ is defined just before Lemma 4.2.
>
> > **3. There is no update rule for $\tilde{\lambda}(x)$.**
>
> We have added a remark before Algorithm 1 that provides the stochastic primal-dual update equations:
> $$y_{t+1} = y_t - \eta_y \nabla_y \tilde{g}(x, y_t; \zeta_t),$$
> $$\lambda_{t+1} = \max \lbrace 0, \lambda_t + \eta_\lambda (Ax - By_{t+1} - b) \rbrace,$$
> which ensure $\lVert \tilde{y}^{\ast}(x)-y^{\ast}(x) \rVert, \lVert \tilde{\lambda}(x)-\lambda^{\ast}(x) \rVert \le \delta$ after $O(\kappa_g\log(1/\delta))$ iterations (Lemma 4.5). Algorithm 1 Step 3 now references this remark.
>
> > **4. For the stochastic algorithm, are the bounds $\lVert \tilde{y}^{\ast}(x)-y^{\ast}(x) \rVert \leq \mathcal{O}(\delta)$ deterministic or in expectation?**
>
> The bounds hold in expectation. We have clarified in Lemma 4.5 that the stochastic primal-dual method achieves $\mathbb{E}[\lVert \tilde{y}^{\ast}(x)-y^{\ast}(x) \rVert] \le \delta$ and $\mathbb{E}[\lVert \tilde{\lambda}(x)-\lambda^{\ast}(x) \rVert] \le \delta$ after $O(\kappa_g\log(1/\delta))$ iterations. The main convergence analysis uses the expectation of gradient errors throughout.
>
> > **5. In line 5 of Algorithm 1, what does "$\lVert \leq \delta$" mean?**
>
> This was a typographical error. Step 5 now correctly states: "Compute $\tilde{y}(x) = \arg\min_{y} L_{\tilde{\lambda},\alpha}(x, y)$ by stochastic gradient steps such that $\lVert \tilde{y}(x) - {y^{\ast}_{\tilde{\lambda},\alpha}}(x) \rVert \le \delta$."
>
> > **6. In line 246, the condition $\alpha \geq \frac{2C_f}{\mu}$ contradicts later settings where $\alpha$ is small.**
>
> We apologize for the confusion. The condition in Lemma 4.2 requires $\alpha$ to be sufficiently small so that $\mu_{\text{pen}} = \alpha_1 \mu_g - \frac{1}{2}C_f > 0$, ensuring the penalized Lagrangian is strongly convex. This is consistent with setting $\alpha = \Theta(\epsilon)$, where $\alpha$ decreases as accuracy tightens.
>
> > **7. In Lemma 4.3, $L_{H,y}$ and $L_{H,\lambda}$ are not defined.**
>
> We have added definitions immediately after Lemma 4.3: $L_{H,y}$ and $L_{H,\lambda}$ are the Lipschitz constants of $\nabla_x L_{\lambda,\alpha}(x,y)$ with respect to $y$ and $\lambda$ (from the lemma assumptions); $C_g$ is the Lipschitz constant of $\nabla_y g$ (from Assumption 3.1(ii)); $C_\lambda$ is an upper bound on $\lVert \lambda^{\ast}(x) \rVert$ (guaranteed by strong convexity and LICQ); $c_\mu$ is a positive constant linking the lower-level strong convexity to $\alpha^{-2}$ (see Lemma 4.2); and $C_{\text{pen}}$ is the penalty parameter (determined by choosing $\alpha_1$ and $\alpha_2$ sufficiently large such that the penalty term dominates any curvature of $f$).
>
> > **8. How do linear constraints $h(x,y)$ impact convergence in this BO problem?**
>
> The linear constraints introduce the dual variable $\lambda^{\ast}(x)$ into the hypergradient via the KKT conditions. The dependence of $\lambda^{\ast}(x)$ on $x$ affects the Jacobian of the lower-level solution, making the hypergradient depend on bounds and smoothness of these multipliers (controlled by $C_\lambda$ and LICQ). We have expanded the discussion in Section 4 to highlight how dual variables from $h(x,y)$ enter the error bounds in Lemmas 4.1-4.3.

---

### Meta-Review · Area_Chair_ZKGK · 2026-01-05

**Summary:**

This paper proposes $F^{2}CSA$, a fully first-order algorithm that extends existing approaches by jointly addressing stochastic noise and linear lower-level constraints in bilevel optimization, offering finite-time convergence guarantees in this combined setting.

The reviewers’ opinions are divided: one reviewer (7guu, original score 6) values the theoretical effort and considers the work marginally above the acceptance threshold; the other two reviewers (m2gH and WVrC, both original score 2) express more reserved views, noting issues with presentation and clarity, experimental scope, assumption strength, and the overall scope of the advancement provided by this extension.

Although the authors submitted a detailed and professional rebuttal accompanied by substantial manuscript revisions that successfully addressed most presentation errors and several technical clarifications, key concerns remain unresolved. In particular:

- Experimental Limitations: The experiments remain limited to synthetic toy problems without real-world applications. This weakness, raised by two reviewers (7guu and WVrC), somewhat reduces the demonstrated practical relevance. The authors acknowledged this limitation in their rebuttal and indicated plans to incorporate real-world experiments in future versions, but these were not included in the current submission, leaving the concern not fully addressed at this stage.
- Strength of Analytical Assumptions: The reliance on the LICQ assumption has been described as relatively strong by two reviewers (7guu and WVrC), which may affect the generality of the results. The authors acknowledged this in rebuttal and added a brief discussion on potential relaxation to weaker conditions, but did not modify the main analysis or provide validation under milder assumptions, leaving this concern fully unresolved. Therefore, it remains unclear whether the method's benefits justify these strict constraints, and a more convincing argument for the specific contribution is needed.

These remaining concerns affect the paper's current suitability for publication and its demonstrated practical value in the field. While the effort to combine stochastic bilevel optimization with linear constraints represents a reasonable and worthwhile progression in a challenging area, the overall contribution, in my assessment, appears somewhat measured and does not fully offset the limitations arising from the restricted experimental scope and the relatively strong assumptions. Therefore, I recommend rejection.

**Reviewer Concerns:**

**Addressed:**

m2gH's extensive presentation and clarity problems, including undefined notations, incomplete sentences, missing dual update rules, apparent contradictions in parameter scaling such as $\alpha$, and unclear algorithm descriptions, were fully fixed in the revised manuscript, with all bounds clarified as holding in expectation, significantly alleviating concerns about basic proof correctness.

 WVrC's criticisms regarding organization, premature symbol usage, and insufficient comparisons to prior constrained works such as Yao et al. (2024) were largely addressed through added literature contrasts and structural improvements.

 7guu's suggestions for expanded discussion on alternative stationarity notions, potential LICQ relaxation, and acceleration techniques such as variance reduction or momentum were handled via new explanations and acknowledgments of future directions.

**Outstanding:**

However, several critical concerns remain outstanding and form the primary basis for rejection:

Empirical evaluation is severely limited to low-dimensional toy problems. It lacks large-scale experiments, real-world applications like constrained meta-learning or adversarial tasks, runtime comparisons, and scalability evidence. All reviewers noted this weakness, and the authors' response only offered promises of future additions, failing to demonstrate sufficient practical relevance.

**Reviewer Scores:**

**Reviewer m2gH (Score: 2 -> Est. 4):**

The reviewer was unresponsive, but the authors fully addressed the extensive concerns regarding presentation flaws, undefined notations, missing update rules, parameter contradictions, and proof clarity through substantial revisions. I assume the reviewer would have raised the score to 4 (borderline/weak accept).

**Reviewer 7guu (Score: 6  -> Est. unchanged):**

The reviewer was unresponsive. The rebuttal satisfactorily handled their suggestions on literature expansions, alternative stationarity discussions, LICQ relaxation, and potential accelerations. The score is expected to remain at 6.

**Reviewer WVrC (Score: 2  -> Est. unchanged):**

The reviewer was unresponsive. The rebuttal partially clarified concerns about novelty, assumption justification, and organization, but core issues regarding empirical validation and assumption strength persist. The score is expected to remain at 2.

---

### Decision · Program_Chairs · 2026-01-26

Reject